# Microcanonical Langevin Ensembles: Advancing the Sampling of Bayesian Neural Networks

**Emanuel Sommer**
Department of Statistics, LMU Munich
Munich Center for Machine Learning (MCML)
Munich, Germany
emanuel@stat.uni-muenchen.de

**Jakob Robnik**
Physics Department
University of California
Berkeley, USA
jakob_robnik@berkeley.edu

**Giorgi Nozadze**
Department of Statistics, LMU Munich
Eraneos Analytics Germany GmbH
Munich, Germany
giorgi.nozadze@eraneos.com

**Uroš Seljak**
Physics Department, University of California
Lawrence Berkeley National Laboratory
Berkeley, USA
useljak@berkeley.edu

**David Rügamer**
Department of Statistics, LMU Munich
Munich Center for Machine Learning (MCML)
Munich, Germany
david@stat.uni-muenchen.de

## Abstract

Despite recent advances, sampling-based inference for Bayesian Neural Networks (BNNs) remains a significant challenge in probabilistic deep learning. While sampling-based approaches do not require a variational distribution assumption, current state-of-the-art samplers still struggle to navigate the complex and highly multimodal posteriors of BNNs. As a consequence, sampling still requires considerably longer inference times than non-Bayesian methods even for small neural networks, despite recent advances in making software implementations more efficient. Besides the difficulty of finding high-probability regions, the time until samplers provide sufficient exploration of these areas remains unpredictable. To tackle these challenges, we introduce an ensembling approach that leverages strategies from optimization and a recently proposed sampler called Microcanonical Langevin Monte Carlo (MCLMC) for efficient, robust and predictable sampling performance. Compared to approaches based on the state-of-the-art No-U-Turn Sampler, our approach delivers substantial speedups up to an order of magnitude, while maintaining or improving predictive performance and uncertainty quantification across diverse tasks and data modalities. The suggested Microcanonical Langevin Ensembles and modifications to MCLMC additionally enhance the method's predictability in resource requirements, facilitating easier parallelization. All in all, the proposed method offers a promising direction for practical, scalable inference for BNNs.

## 1 Introduction and related literature

Sampling-based inference for Bayesian Neural Networks (BNNs) has garnered significant interest as a principled approach to addressing the analytically intractable challenge of probabilistic deep learning (Izmailov et al., 2021; Wiese et al., 2023; Papamarkou et al., 2024). New methods, such as subspace inference (Izmailov et al., 2020; Dold et al., 2024), are being explored alongside emerging applications in diverse domains where effective uncertainty quantification is crucial, including

healthcare (Peng et al., 2020) and physics (Cranmer et al., 2021).Papamarkou et al. (2022); Sommer et al. (2024) highlight several shortcomings in current sampling-based approaches, particularly the need for proper initialization of sampling procedures and the challenge of capturing multimodality.

**Samplers and problem setup** Hamiltonian Monte Carlo (HMC; Duane et al., 1987) and Underdamped Langevin Monte Carlo (Leimkuhler & Reich, 2009) are the gold standard algorithms for high-dimensional sampling problems. However, their performance is known to be sensitive to their hyperparameters, such as the step size, preconditioning, and the momentum decoherence rate (Neal, 2011). They are therefore combined with an automatic hyperparameter adaptation algorithm. Several ensemble-based schemes have been developed in recent years (Sountsov & Hoffman, 2022; Hoffman & Sountsov, 2022; Riou-Durand et al., 2023), but all of these schemes critically depend on the ensemble variance of the parameters to tune the critical momentum decoherence rate hyperparameter and in some cases also the (preconditioned) step size. As such, they are not applicable to the highly multimodal posteriors of BNNs.

In the context of gradient-free sampling, a variety of ensembled algorithms have been developed, for example, Preconditioned Monte Carlo (pocoMC; Karamanis et al., 2022a;b) , Nested Sampling (Skilling, 2004) and Elliptical Slice Sampling (Murray et al., 2010). These algorithms address the multimodality problem; however, they scale poorly to high-dimensional settings. For example, Eliptical Slice Sampling is frequently used for BNNs (Izmailov et al., 2020; Dold et al., 2024) for sampling in a small subspace (e.g., 2- or 3-dimensional) of the parameter space, but cannot be scaled to much larger dimensions. While alternative methods for multimodal sampling exist, they are also not well-suited for the high-dimensional and highly multimodal BNN setting. For instance, stochastic localization methods, like those proposed by Grenioux et al. (2024), can handle moderate multimodality but are sensitive to hyperparameters (like the integration initialization $t_0$ for stochastic localization) and lack scalability. Similarly, the Liouville Flow Importance Sampler (Tian et al., 2024) offers unbiased sampling but requires the training of a neural flow model, which needs to become increasingly complex as the dimensionality and complexity of the problem grows, limiting its scalability. The same applies to the learned vector field in the path-guided particle-based method of Fan et al. (2024), with an empirical comparison provided in Appendix A.1.3.

**HMC and NUTS** It is therefore not surprising that a sequential HMC variant, the No U-turn Sampler (NUTS; Hoffman & Gelman, 2014), is still viewed as "[t]he only MCMC algorithm that theoretically scales to high dimensions across a broad class of models" (Štrumbelj et al., 2024). While the scalability of HMC allows the application to the full parameter space of moderately-sized neural networks and its NUTS variant provides an (almost) tuning-free approach of HMC, it cannot sufficiently explore the many modes present in BNN posteriors(Wiese et al., 2023). The recent HMC variant Symmetric Split HMC (Cobb & Jalaian, 2021) improves HMC's memory scalability but comes with other drawbacks, including sensitive hyperparameters. We provide an empirical comparison and further discussion in Appendix A.1.3. Standard priors, like isotropic Gaussians, can further lead to initialization issues, where samplers start in low-probability regions, resulting in slow convergence or samplers getting stuck. To address these issues, Sommer et al. (2024) propose a Bayesian Deep Ensemble (BDE), an ensemble of many short warmstarted Markov chain Monte Carlo (MCMC) chains. With this, the authors increase the exploration capability and are able to achieve state-of-the-art predictive and uncertainty quantification (UQ) performance on common benchmark tasks. Although much faster than previously employed methods by leveraging parallelization and efficient implementations, Sommer et al. (2024) rely on NUTS. This imposes a significant computational burden and results in a method where the sampling phase still dominates the computational costs by far.

**Our contributions** In an effort to improve the scaling and efficiency of sampling-based inference for BNNs, we identified Microcanonical Langevin Monte Carlo (MCLMC; Robnik et al., 2023) as a possible alternative MCMC-based solution. In recent experiments, the authors were able to show that MCLMC can be computationally superior to NUTS while providing the same quality of samples in downstream metrics. While MCLMC demonstrates computational advantages over NUTS in unimodal and sequential sampling tasks, it cannot address the multimodality, numerical instability, and scalability challenges of BNNs without significant modifications (e.g. see Appendix A.1.2). We embed adapted MCLMC as the key ingredient in our approach, including deep ensemble initialization for enhanced exploration, adjustments to ensure numerical stability in high-dimensional settings,

and optimizations targeting critical bottlenecks. The resulting method, Microcanonical Langevin Ensemble (MILE), comes with automated tuning and works reliably out of the box. Extensive experiments highlight that MILE achieves state-of-the-art performance while being up to an order of magnitude faster than previous sampling-based approaches.

## 2 BACKGROUND

In this work, we denote neural networks with $f : \mathcal{X} \to \mathcal{Y}, \mathcal{X} \subseteq \mathbb{R}^p, \mathcal{Y} \subseteq \mathbb{R}^m$. We parameterize the network with $\boldsymbol{\theta} \in \Theta \subseteq \mathbb{R}^d$, denoting the vector of all flattened and concatenated weights and biases. To express the epistemic uncertainty about $\boldsymbol{\theta}$, we treat $\boldsymbol{\theta}$ as a random variable with prior density $p(\boldsymbol{\theta})$ and denote its posterior density as $p(\boldsymbol{\theta}|\mathcal{D}) = p(\mathcal{D}|\boldsymbol{\theta})p(\boldsymbol{\theta})/p(\mathcal{D})$, with observed data $\mathcal{D} \in (\mathcal{X} \times \mathcal{Y})^n$. We will assume a standard isotropic unit variance Gaussian prior $\mathcal{N}(0, I)$ if not specified otherwise. A more detailed analysis of the prior influence is beyond the scope of this work. The posterior predictive density (PPD) quantifies the uncertainty of predicting unseen labels $\boldsymbol{y}^* \in \mathcal{Y}$ given features $\boldsymbol{x}^* \in \mathcal{X}$ by integrating over the posterior distribution of the parameters $\boldsymbol{\theta}$:

$$p(\boldsymbol{y}^*|\boldsymbol{x}^*, \mathcal{D}) = \int_{\Theta} p(\boldsymbol{y}^*|\boldsymbol{x}^*, \boldsymbol{\theta})p(\boldsymbol{\theta}|\mathcal{D}) \, \mathrm{d}\boldsymbol{\theta}. \tag{1}$$

### 2.1 MONTE CARLO SAMPLING

In the setting of sampling-based inference, we estimate the analytically intractable integral in Eq. (1) using samples from the posterior distribution with density $p(\boldsymbol{\theta}|\mathcal{D})$. These are obtained by Markov chain Monte Carlo (MCMC) methods, which construct a Markov chain whose stationary distribution is the posterior distribution or close to it. In practice, we gather these samples from $K$ independent chains, each with $S$ samples, yielding the set $\{\boldsymbol{\theta}^{(k,s)}|k \in [K], s \in [S]\}$ based on which we can perform prediction and uncertainty quantification (Andrieu et al., 2003; Gelman et al., 2013). The approximation of Eq. (1) then has the form

$$p(\boldsymbol{y}^*|\boldsymbol{x}^*, \mathcal{D}) \approx \frac{1}{K \cdot S} \sum_{k=1}^{K} \sum_{s=1}^{S} p\left(\boldsymbol{y}^*|\boldsymbol{x}^*, \boldsymbol{\theta}^{(k,s)}\right). \tag{2}$$

**Error analysis and Metropolis-Hastings adjustment** The error of this approximation is composed of three terms: initialization error, discretization error, and Monte Carlo error. The initialization error is a transient effect caused by the chains' ensemble distribution having not yet reached the stationary distribution (burn-in phase). The Monte Carlo error is the variance caused by the finite number of samples $S$ and chains $K$. The discretization error is caused by the finite step size used to numerically simulate the sampler's dynamics. This causes $p(\boldsymbol{\theta}|\mathcal{D})$ to no longer be the stationary distribution. The Metropolis-Hastings (MH) scheme is typically employed to completely eliminate the discretization error, but this comes at an expense of shorter step size. This is because the acceptance rate depends exponentially on the squared energy error, which grows linearly with dimensionality. Therefore, the step size needs to be decreased as the number of parameters increases in order to maintain a fixed acceptance rate, causing the sampler to move more slowly in higher dimensions. The MH algorithm is further prone to degeneration of the acceptance rate if the chains are initialized from a distribution, which is not already very close to the stationary distribution, causing slow convergence and large initialization error (Durmus & Eberle, 2023). Without the Metropolis adjustment on the other hand, the discretization error depends strongly on the step size. Hence, slightly reducing the step size causes the discretization error to become negligible compared to the initialization and Monte Carlo error. Given these arguments, we choose to omit the MH adjustment in our method, relying instead on careful initialization and step size tuning to manage the error terms.

### 2.2 MICROCANONICAL LANGEVIN MONTE CARLO

A recently proposed sampling method outside the BNN research field is Microcanonical Langevin Monte Carlo (MCLMC). The time evolution of the MCLMC sampler (Robnik & Seljak, 2024) is governed by the stochastic differential equation

$$\mathrm{d}\boldsymbol{\theta} = \boldsymbol{u} \, \mathrm{d}t, \qquad \mathrm{d}\boldsymbol{u} = \left(1 - \boldsymbol{u}\boldsymbol{u}^\top\right)\left((d-1)^{-1}\nabla \log p(\boldsymbol{\theta}|\mathcal{D}) \, \mathrm{d}t + \eta \, \mathrm{d}\boldsymbol{W}\right), \tag{3}$$

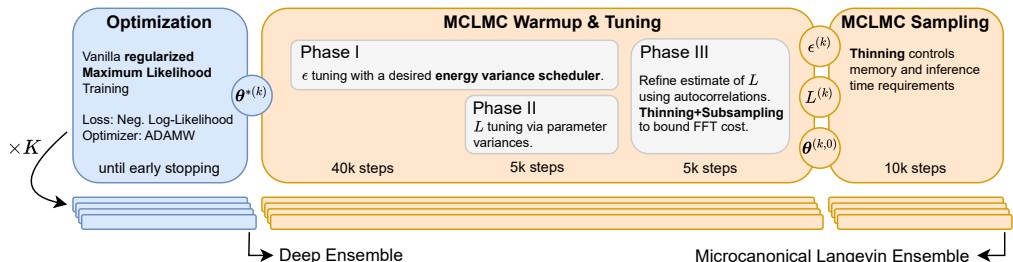

Figure 1: Flowchart illustrating our proposed procedure for obtaining a Microcanonical Langevin Ensemble (MILE) for BNNs. The process involves three main stages: optimization, MCLMC warmup and tuning, and MCLMC sampling. These steps are parallelized to generate an ensemble of $K$ members. The number of MCLMC steps for each tuning phase and the final sampling phase are annotated, and carryovers between stages are highlighted in circles.

where $\boldsymbol{W}$ is the Wiener process and $\eta$ is a free parameter equivalent to the typical length traveled in the parameter space before momentum decoherence, often denoted with $L$. Robnik & Seljak (2024) use the drift-diffusion discretization to numerically solve Eq. (3). The drift part (Eq. (3) without the last, stochastic term) is solved by the minimal norm integrator (Takaishi & de Forcrand, 2006; Omelyan & Kovalenko, 2013). In the limit of small step size $\epsilon$, the resulting Markov chain has $p(\boldsymbol{\theta}|\mathcal{D})$ as a stationary distribution (Robnik & Seljak, 2024). Smaller step size results in a smaller discretization error but increases the Monte Carlo and initialization errors because the sampler is moving more slowly. We therefore wish to reduce the step size to the point where the discretization error becomes smaller than the other two sources of error, but not much further. Robnik et al. (2023) propose the energy error variance per dimension (EEVPD) as a measure of the discretization error and show that one can control the discretization error by controlling EEVPD. Analogously to adapting the step size to match some targeted acceptance rate in Metropolis-adjusted algorithms, one can adapt the step size to match some desired EEVPD in the unadjusted algorithms. This inexact but highly efficient approach with bias control is the one we will also take in our work.

Note that the stationary distribution is independent of the free parameter $\eta$ and in particular also holds for the deterministic dynamics, $\eta = 0$. This is not the case in HMC, where momentum resampling is essential to maintain the desired stationary distribution. Being more deterministic allows MCLMC to converge faster during the exploitation phase.

The parameter $\eta$ is still important, as it controls the rate of the momentum decoherence and forces the dynamics to move to the unexplored parts of the parameter space. Robnik et al. (2023) found that a good performance is achieved when $L/\epsilon$ is on the order of the chain's autocorrelation time. As the autocorrelation time is quite expensive to evaluate, they also propose to set $L$ to the size of the posterior modes, by computing the variance of the parameters.

Based on these considerations, they propose a three-stage tuning scheme, which we now refer to as the three phases:

1. Phase I: Adapt the step size to match the desired EEVPD and complete the burn-in.
2. Phase II: Estimate parameter variance to obtain an initial estimate for $L$.
3. Phase III: Estimate autocorrelation time to refine the estimate of $L$.

## 3  MICROCANONICAL LANGEVIN ENSEMBLES

In order to embed MCLMC in a robust sampling pipeline (cf. Fig. 1) that works effectively for BNNs and exploits the idea of ensembling, we will discuss the combination of optimization techniques and sampling as well as various aspects of required MCLMC modification in the following. Without these modifications, MCLMC struggles with exploration and exhibits high failure rates, particularly in high-dimensional settings (see Appendix A.1.2).

### 3.1  ENSEMBLING FOR REDUCED INITIALIZATION ERROR

As discussed in Section 2.1, the error in the approximation of the predictive posterior can be decomposed into three parts. While the initialization error is described in Section 2.1 as a "transient

effect", this error should not be mistakenly classified as unimportant, and treating it with special care is particularly important in the application of BNNs. Similar to most other samplers, it is not unlikely for MCLMC to get stuck in low-probability regions of the high-dimensional and highly complex posterior surfaces of BNNs. Our proposed solution is therefore to combine sampling with an optimization step (Fig. 1, blue part). To prevent chains in the sampling step from being initialized in these unfavorable regions, we run $K$ optimization steps to obtain starting values $\boldsymbol{\theta}^{*(k)}, k \in [K]$ for each of the $K$ chains. As a byproduct, we obtain a deep ensemble (DE; Lakshminarayanan et al., 2017) with $K$ members.[1] Similar to Sommer et al. (2024), we found that using optimized neural networks as starting values can reduce this error notably and prevents the sampler from getting stuck.

## 3.2 TUNING PHASE ADAPTIONS

In the previous subsection, we proposed to optimize starting values when using MCLMC by running an ensemble of MCLMC chains. Optimizing the starting values of each chain is, however, not sufficient to make MCLMC work numerically stable and efficiently for BNNs (see Appendix A.1.2). We therefore discuss three components of MCLMC's tuning phase requiring further adaptions in applications to BNNs, and propose scaling related modifications to phase III of MCLMC.

**Step size**  Robnik et al. (2023) suggest to initialize the step size $\epsilon$ with $\sqrt{d}$. While this default works well for their applications, this would result in too large values even for rather small neural networks. Too large step sizes, in turn, will result in significant changes of the energy and thereby introduce serious numerical problems. As MILE initializes chains already in a region of high probability, it is reasonable to reduce $\epsilon$ notably. In practice, this means we can set the step size to the optimizer's learning rate used in the optimization step of MILE. Besides reducing the initial discretization error, this also ensures a more localized exploration early in the burn-in phase and close to the optimized solution while allowing for larger steps as tuning progresses (which can also be confirmed empirically).

**Energy variance scheduler**  Another tuning parameter in phase I of MILE is the energy variance scheduler allowing control of the trade-off between exploitation and exploration. Having already optimized $K$ starting values, MILE does not require an excessive amount of exploitation but each MCLMC ideally focuses more on exploitation. In contrast to the tuning scheme proposed in Robnik et al. (2023), we do not employ a fixed desired energy variance level but use a linear scheduler starting with a higher desired energy variance and gradually decreasing it. By doing this we foster noisier exploration at the beginning of the warmup phase and increase the exploitation towards the end of it. Doing this in parallel across an ensemble of chains, our idea resembles the popular strategy of using cyclical learning rates in SG-MCMC sampling (Zhang et al., 2020). But rather than going through various exploration and exploitation phases we do one exploration-exploitation cycle per chain and thus parallelize this idea.[2]

**Effective sample size**  MCLMC further requires setting a desired effective sample size (ESS) level within the EEVPD estimation phase. While this level is generally a choice of the practitioner and the available computational resources, one option is to set the ESS to 10% of the total number of posterior samples. As previous approaches working with NUTS also report an ESS of around 10% of the total number of samples, this value ensures a sample quality that is as good as the one of NUTS.

**Phase III bottleneck**  When scaling the MCLMC tuning algorithm to higher dimensions, a computational bottleneck of phase III is to estimate the empirical ESS, which is done with a Fast Fourier Transform (FFT, Bracewell & Bracewell, 1986). The FFT is computed for each parameter across all samples. Although fast implementations scale with $\mathcal{O}(S \log S)$, this can severely impact the runtime for increased dimensions and amount of samples. We therefore propose to subsample parameters

---

[1]Note that the optimization of these $K$ models actually has negligible costs compared to the costs of current state-of-the-art sampling approaches.

[2]We empirically found that the linear schedule from 0.5 to 0.1 worked well in a variety of settings. We also explored exponentially decaying schedulers starting with high desired energy variance, but these proved to be empirically inferior to the aforementioned linear schedule. This is in line with the intuition that excessive exploration is not necessary due to the DE initialization.

if the dimensionality is $d > 2000$, which linearly decreases the runtime. We also suggest applying thinning to the samples such that the number of samples used for the FFT is bounded by $10e3$.

### 3.3 EFFECTIVE COMPUTATIONAL BUDGET ALLOCATION

The third error caused by the finite number of samples and chains is the Monte Carlo error. Unlike NUTS, which varies the number of leapfrog steps for each proposal and requires one gradient evaluation per leapfrog step, MCLMC employs a deterministic approach that requires two gradient evaluations per sample due to the use of a minimal norm integrator. This structure makes MCLMC's computational requirements much more predictable, offering a significant practical advantage. In contrast, the number of steps taken by NUTS can exhibit substantial variance both within and across tasks, leading practitioners to set an upper limit on leapfrog steps. With better foreseeable computational requirements, we can optimize the balance between the number of chains and the number of samples in MILE in order to minimize the Monte Carlo error for a given computational budget. As recent literature suggests that MCLMC can be much more efficient than NUTS in obtaining effective samples in simple problems, we propose to collect a relatively small number of samples from each chain and thereby limit computational resources spent on exploring individual local modes by MCLMC but relying on DE for exploration.

**Warmup stage**   It is not advisable to directly start chains from the DE optimized points, as these modes do not coincide with the so-called typical sets (Betancourt, 2018), we suggest allocating 40k steps, i.e., 80k gradient evaluations for MILE's warmup phase. Compared to NUTS, which often requires more than 90k leapfrog steps/gradient evaluations for warmup in BNN tasks, this is a conservative lower bound. This is followed by the two shorter phases, II and III, where we allocate 5k steps each to ensure robust estimations of the momentum decoherence scale $L$.

**Sampling stage**   As MCLMC samples exhibit significant autocorrelation, we can further reduce memory costs by applying thinning without notable reduction in sample quality while also not increasing inference time as long as the costs of thinning remain negligible compared to the overall sampling time. In our setup, after 50k warmup steps, we propose using 10k steps for the sampling phase. The choice of thinning interval is then based on the memory and inference time constraints, i.e., the posterior sample budget.

In this work, we explore two scenarios: one with a budget of 1k samples per chain and another with 100 samples, corresponding to thinning intervals of 10 and 100, respectively. Overall, we propose a fixed budget of 60k steps per chain, providing a predictable sampling process. By adjusting thinning appropriately, one can further effectively manage memory and inference time requirements.

## 4 EXPERIMENTS

In this section, we evaluate the feasibility of applying MILE to sampling-based inference for BNNs.

- **Datasets and models**: We replicate the benchmark by Sommer et al. (2024) but also extend it to other datasets (Ionosphere, Income, IMDB, MNIST, F-MNIST) and models (convolutional and attention-based neural networks).

- **Methods**: As in Sommer et al. (2024), we investigate the improvement of our proposed approach over a DE, but also compare against the current state-of-the-art BDE approach based on the NUTS sampler.

- **Runtime comparisons**: Following this, we conduct a series of ablation studies, carefully examining how MILE scales compared to BDE.

- **Tuning and hyperparameters**: We finally validate the robustness of MILE's hyperparameters, supporting our claim that MILE is an auto-tuned off-the-shelf procedure like NUTS.

Details on the experimental setting and the implementation can be found in Appendix A.2. The diagnostics and evaluation metrics employed are detailed in Appendices A.3 and A.4.

Table 1: Average hold-out LPPD and RMSE performance as well as wallclock runtime of the DE baseline, BDEs and MILE, respectively, for the six datasets (in different rows) over 3 data splits. A table including standard deviations can be found in Appendix A.1. The wallclock times of the samplers represent the additional sampling time on top of the DE fit which is also reported.

| | LPPD (↑) | | | RMSE (↓) | | | Time (min) | | |
| --- | --- | --- | --- | --- | --- | --- | --- | --- | --- |
| | DE | BDE | MILE | DE | BDE | MILE | DE | BDE | MILE |
| A | 0.024 | 0.558 | **0.612** | 0.309 | **0.214** | **0.206** | 0.62 | 2.25 | **0.84** |
| B | 0.390 | 0.625 | **0.645** | 0.251 | **0.242** | **0.236** | 5.67 | 48.29 | **5.40** |
| C | −0.072 | **0.301** | 0.336 | 0.304 | **0.273** | **0.250** | 0.33 | 1.56 | **0.77** |
| E | 1.227 | 2.072 | **2.300** | 0.120 | 0.045 | **0.034** | 0.39 | 1.11 | **0.75** |
| P | −1.024 | **−0.760** | **−0.750** | 0.742 | **0.703** | **0.702** | 12.37 | 152.85 | **19.50** |
| Y | 1.623 | **2.674** | 2.859 | 0.081 | 0.083 | **0.033** | 0.16 | **0.58** | 0.64 |

## 4.1 BENCHMARKS

The goal of this section is to demonstrate 1) the feasibility of MILE in BNN settings, 2) highlight its superior predictive performance measured using the root mean squared error (RMSE) or Accuracy, its improved UQ measured using the LPPD, and 3) its improvements in runtime.

### 4.1.1 UCI BENCHMARKS

Using the six UCI datasets previously analyzed in the context of sampling-based inference in Sommer et al. (2024) we replicate their benchmark using a ReLU network with two hidden layers and 16 neurons each (see Appendix A.2 for further details). We compare the predictive accuracy, UQ, and runtime of state-of-the-art BDE and the DE baselines against MILE, configured as described in Section 3.

**Performance results** Table 1 shows the results, indicating that MILE consistently matches or outperforms the other methods in predictive performance and UQ while significantly reducing the computational cost. This is especially evident in larger datasets like `bikesharing` (B) and `protein` (P), where MILE achieves sampling speeds nearly ten times faster than BDE. In most cases, the additional MCLMC sampling after DE optimization is as fast as the DE fitting phase itself. It is noteworthy that this is a big step for sampling-based inference, yielding a time complexity comparable to DE ($\approx 2\times$runtime), while providing better and more principled uncertainty measures.

**Resource predictability** As elaborated before, MILE's efficiency is driven by its much fewer gradient evaluations and predictable runtime. Unlike BDE, where runtime is highly variable in the leapfrog steps taken by NUTS, MILE's fixed number of steps allows easy runtime forecasts once the cost of a single gradient evaluation is known. This is especially crucial for expensive sampling task such as the posterior sampling of BNNs. Fig. 2 illustrates the significant variability in BDE's computational cost for generating 1000 posterior samples, both within and across tasks. This variability often exceeds the entire budget for sampling in MILE. Moreover, in parallel settings,

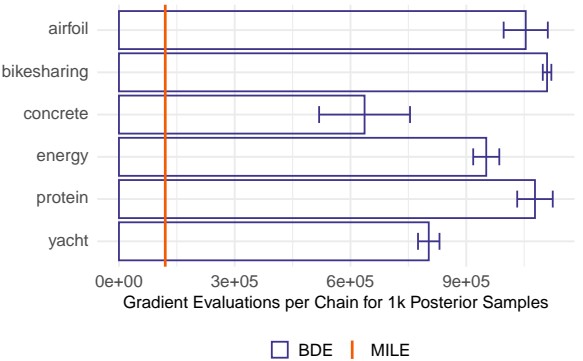

Figure 2: Average gradient evaluations per chain for 1000 posterior samples for the experiments reported in Table 1.

the BDE's performance is bottlenecked by the most expensive chain, while MILE's deterministic nature ensures perfect load balancing for concurrent sampling (see Section 3.3).

**Diagnostics** For diagnostic purposes and to evaluate the quality of samples drawn by MILE and BDE, we evaluate the effective sample size (ESS, Vehtari et al., 2021) and the chainwise 4-split $\widehat{cR}$ (Sommer et al., 2024). As can be seen in Figures 6a and 6b, MILE also outperforms BDE in the sampling and local mixing quality, revealing higher average ESS and smaller $\widehat{cR}$ values.

### 4.1.2 EXTENDED BENCHMARKS

In addition to existing sampling-based inference benchmarks, we extend our analysis to more complex and larger models, showcasing the successful application of MILE to convolutional neural networks (CNNs), classification tasks, and sequential models across different data modalities (see Appendix A.2 for further details). The main reason we are able to advance existing benchmarks is the feasibility of sampling such tasks using MILE. As it would have taken weeks to obtain the inference using NUTS, we compare MILE mainly to the DE baseline but provide results for NUTS for smaller models in the Appendix. Additionally, we report the average performance of a single DE member and individual MCMC chain, which highlights the additional benefits of ensembling.

**Results** The results, presented in Table 2, not only confirm the accuracy and UQ improvements over the DE baseline but also demonstrate performance gains in much larger models and new problem domains, confirming the positive effect of the additional exploration step taken by MILE.

Table 2: Hold-out test performance of MILE and baselines on various classification tasks using fully-connected (FCN), convolutional (CNN) and attention-based (ATT) networks. ATT (v2, v3) employ pretrained embeddings.

| Dataset | Model | # Params | Accuracy ($\uparrow$) | | | | LPPD ($\uparrow$) | | | |
| --- | --- | --- | --- | --- | --- | --- | --- | --- | --- | --- |
| | | | Avg. Single | | Ensemble | | Avg. Single | | Ensemble | |
| | | | DNN | Chain | DE | MILE | DNN | Chain | DE | MILE |
| Ionosphere | FCN (v1) | 850 | 0.930 | **0.958** | **0.958** | **0.958** | -0.404 | -0.168 | -0.309 | **-0.167** |
| Income | FCN (v2) | 2386 | 0.843 | **0.851** | 0.846 | **0.851** | -0.334 | -0.315 | -0.318 | **-0.313** |
| IMDB | ATT (v1) | 68448 | 0.715 | 0.714 | **0.718** | 0.717 | -0.550 | -0.546 | **-0.544** | **-0.544** |
| IMDB | ATT (v2) | 55778 | 0.779 | 0.754 | 0.781 | **0.782** | -0.591 | -0.583 | -0.562 | **-0.481** |
| IMDB | ATT (v3) | 106190 | 0.786 | **0.790** | 0.786 | 0.788 | -0.509 | -0.493 | -0.507 | **-0.491** |
| MNIST | CNN (v1) | 7452 | 0.916 | 0.939 | 0.956 | **0.970** | -0.299 | -0.209 | -0.179 | **-0.129** |
| F-MNIST | CNN (v1) | 7452 | 0.742 | 0.767 | 0.863 | **0.885** | -0.725 | -0.684 | -0.486 | **-0.430** |
| F-MNIST | CNN (v2) | 61706 | 0.890 | 0.919 | 0.918 | **0.925** | -0.361 | -0.225 | -0.227 | **-0.216** |

### 4.2 ABLATION MODEL COMPLEXITY & RUNTIME

The UCI benchmarks have clearly showed the runtime advantage of MILE over BDE. To further investigate the impact of model complexity (number of parameters) and dataset size on these runtimes, we conduct two ablation studies.

**Scaling in model complexity** Fig. 3 shows the evolution of the required sampling time and performance metrics for increased model complexity. For the 5,426-dimensional case, MILE completes sampling in under 30 minutes, while BDE takes several hours. More strikingly, MILE not only remains faster but its performance gap to BDE increases with higher model complexity, indicating both significant efficiency gains and superior performance in higher dimensions. To quantify the time complexity, we fit a linearized power-law model to obtain the expected sampling time with growing $d$: $\mathbb{E}[\log(t_{\text{Sampler}})] = \beta_0 + \beta_1 \log(d)$. Both fits explain almost all the variance in the data with $R^2$ statistics of 0.98 for MILE and 0.96 for BDE, confirming the robustness of the speed advantage. The model fits are illustrated as dashed lines in Fig. 3.

**Scaling in dataset size** We analyze the impact of increasing dataset size. Fig. 3 (top right) illustrates how sampling time evolves for a fixed task and neural network across different data subsets. Using the linear model $\mathbb{E}[t_{\text{Sampler}}] = \beta_0 + \beta_1 n^2$, we obtain fits that again almost perfectly resemble the given data points, with $R^2$ values of 0.99 for MILE and 0.96 for BDE. The ratio of the $\beta_1$ coefficients, 6.6, again highlights MILE's superior scaling in $n^2$. Notably, visual inspection suggests that BDE may scale worse than quadratically, but for consistency with MILE, we apply a conservative quadratic fit. The scaling behavior again closely resembles a power-law and the observed quadratic trend in $n$ likely stems from memory-related hardware slowdowns as dataset sizes increase.

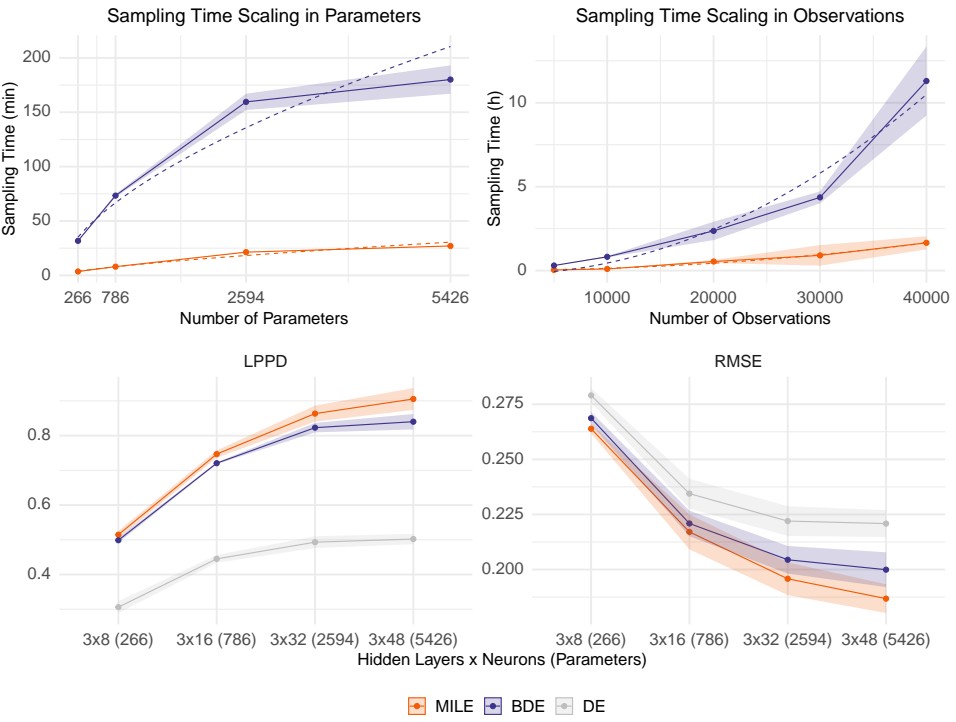

Figure 3: Average sampling wallclock times (minutes, y-axis) of BDE (blue) and MILE (orange) for the `bikesharing` dataset across 4 NN architectures with increasing parameter count (x-axis) on the upper left. Average sampling wallclock times (hours, y-axis) for the `protein` dataset across varying training data sizes (x-axis) on the upper right. Dashed lines indicate power-law and quadratic model fits respectively. In both cases the sampling time ratio between BDE and MILE is around 7-9, independently of the number of parameters and observations. This is a result of NUTS always being close to its maximum number of iterations per sample, which we set to the default value of 1024 gradient calls. It therefore uses around $1024 \times (1000 + 100) \approx 11 \times 10^5$ gradient calls, as displayed also in Figure 2. MILE on the other hand always uses $2 \times 60000 = 12 \times 10^4$ calls, which gives a ratio of 9.2. The bottom row shows hold-out metric performances across 4 network architectures. DE performance for the LPPD and RMSE metrics is indicated as a grey reference. All charts come with standard errors over 3 data splits.

## 4.3 ABLATION HYPERPARAMETER ROBUSTNESS

To substantiate the claim that our proposed MCLMC configuration in MILE is, in fact, a robust and auto-tuned off-the-shelf method, we conducted a series of ablation studies by changing the default hyperparameters of MILE as suggested in Section 3 and examine the impact of these changes on performance. For each ablation, we systematically evaluate the robustness of the approach by changing a single hyperparameter on an appropriate grid. We report both the impact of hyperparameter variations on hold-out test performance metrics and the corresponding tuned values of the sampler's key parameters—the momentum decorrelation scale ($L$) and step size ($\epsilon$). While the effect on the latter ones is interesting the primary focus is to attain estimates of the two parameters that yield robust and good performance. For comparison, we also present the performance of BDE and the DE baseline.

**Results** The results are summarized in Fig. 4. Across all cases, MILE consistently shows robust performance, with minimal sensitivity to changes in the hyperparameters. The changes in the major parameters $L$ and $\epsilon$ are generally small and with such marginal changes -especially to the important $\epsilon$- do not significantly affect the overall performance. An exception is the warmup budget, where slightly improved performance can be observed with a significantly extended warmup phase. However, the improvements are marginal and come at a linear increase in runtime, making the additional computational cost unjustifiable.

In conclusion, these results support the claim that MILE is effectively tuning-free, as in the considered cases, the method's performance does not change drastically with different hyperparameters and is close to optimal under the default settings. This robustness ensures that practitioners can confidently apply the method without extensive tuning, further enhancing its practical utility.

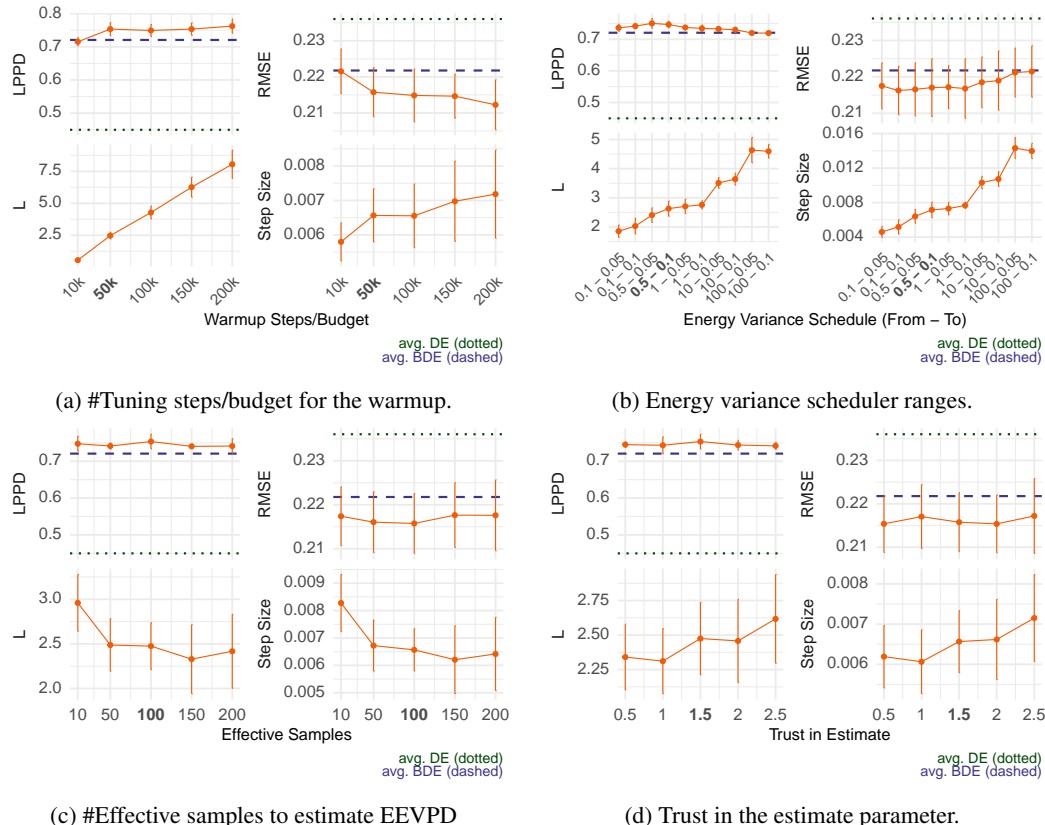

(a) #Tuning steps/budget for the warmup.

(b) Energy variance scheduler ranges.

(c) #Effective samples to estimate EEVPD

(d) Trust in the estimate parameter.

Figure 4: Results of the ablation studies conducted on the `bikesharing` dataset for the robustness of the MILE algorithm to its tuning parameters (x-axes, proposed defaults bold). Both the average hold-out RMSE and LPPD are reported with their standard error for 3 data splits. The same holds for the major parameters of the sampling kernel $L$ and the step size that were tuned by the proposed tuning. "#Effective samples to estimate EEVPD" and "Trust in the estimate" are minor parameters of the step size adaptation algorithm which determine the sample weighting during the EEVPD computation.

## 5 DISCUSSION

In this work, we proposed and evaluated Microcanonical Langevin Ensembles (MILE) for sampling-based inference of Bayesian neural network posteriors. By adapting the recently proposed MCLMC sampler and combining it with starting values obtained via Deep Ensembles, our comparative analysis reveals substantial advancements in performance, sample quality, UQ, and runtime compared to a NUTS-based alternative. Furthermore, the method enhances the predictability of resource requirements due to its deterministic number of gradient evaluations, which also simplifies parallelization. In conclusion, our proposed method can be considered a reliable and efficient off-the-shelf method and thus a big step forward toward making sampling-based inference feasible for generic BNNs.

**Scope of this work and limitations**  While also yielding significant runtime savings for increased dataset dimensions, the goal of this work was to overcome the most prevalent bottlenecks of NUTS-based ensembling—its unfavorable scaling with respect to the number of parameters and unforeseeable resource allocation due its variable and high number of gradient evaluations. As MCLMC and our modifications seem to have solved these problems, a next step for future work that we did not compare in this study is the transition to Stochastic Gradient sampler variants. This transition is typically not straightforward but would overcome another remaining limitation of sampling-based inference, namely the scaling for large-scale datasets. Another possible enhancement of MILE we did not investigate in this work is the use of alternative priors, e.g., discussed in Fortuin et al. (2022).

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

# A APPENDIX

## A.1 FURTHER RESULTS

### A.1.1 BENCHMARKS

**UCI benchmark** The results for the UCI benchmark including standard deviations are given in Table 3.

Table 3: Average hold-out and standard error of LPPD and RMSE performance as well as wallclock time of the DE baseline, BDEs and MILE for the six datasets used in Sommer et al. (2024) over 3 data splits. The wallclock times of the samplers represent the additional sampling time on top of the DE fit which is also reported.

| | LPPD (↑) | | | RMSE (↓) | | | Time (min) | | |
|---|---|---|---|---|---|---|---|---|---|
| | DE | BDE | MILE | DE | BDE | MILE | DE (∀Methods) | BDE | MILE |
| A | $0.024 \pm 0.024$ | $0.558 \pm 0.034$ | $\textbf{0.612} \pm \textbf{0.011}$ | $0.309 \pm 0.005$ | $\textbf{0.214} \pm \textbf{0.013}$ | $\textbf{0.206} \pm \textbf{0.018}$ | $0.62 \pm 0.15$ | $2.25 \pm 0.01$ | $\textbf{0.84} \pm \textbf{0.06}$ |
| B | $0.390 \pm 0.010$ | $0.625 \pm 0.004$ | $\textbf{0.645} \pm \textbf{0.10}$ | $0.251 \pm 0.004$ | $0.242 \pm 0.005$ | $\textbf{0.236} \pm \textbf{0.004}$ | $5.67 \pm 0.98$ | $48.29 \pm 0.51$ | $\textbf{5.40} \pm \textbf{0.02}$ |
| C | $-0.072 \pm 0.018$ | $\textbf{0.301} \pm \textbf{0.088}$ | $\textbf{0.336} \pm \textbf{0.054}$ | $0.304 \pm 0.007$ | $\textbf{0.273} \pm \textbf{0.012}$ | $\textbf{0.250} \pm \textbf{0.014}$ | $0.33 \pm 0.03$ | $1.56 \pm 0.05$ | $\textbf{0.77} \pm \textbf{0.06}$ |
| E | $1.227 \pm 0.037$ | $2.072 \pm 0.036$ | $\textbf{2.300} \pm \textbf{0.066}$ | $0.120 \pm 0.022$ | $0.045 \pm 0.003$ | $\textbf{0.034} \pm \textbf{0.007}$ | $0.39 \pm 0.05$ | $1.11 \pm 0.00$ | $\textbf{0.75} \pm \textbf{0.03}$ |
| P | $-1.024 \pm 0.029$ | $-0.760 \pm 0.010$ | $-0.750 \pm 0.018$ | $0.742 \pm 0.011$ | $\textbf{0.703} \pm \textbf{0.003}$ | $\textbf{0.702} \pm \textbf{0.008}$ | $12.37 \pm 1.00$ | $152.85 \pm 7.80$ | $\textbf{19.50} \pm \textbf{0.01}$ |
| Y | $1.623 \pm 0.101$ | $\textbf{2.674} \pm \textbf{0.216}$ | $\textbf{2.859} \pm \textbf{0.199}$ | $0.081 \pm 0.038$ | $0.083 \pm 0.011$ | $\textbf{0.033} \pm \textbf{0.011}$ | $0.16 \pm 0.01$ | $\textbf{0.58} \pm \textbf{0.01}$ | $0.64 \pm 0.02$ |

**Chain variances** Our analyses of between- and within-chain variances (Fig. 6) show a distinctive pattern of an increasing within-chain variance in layers further away from the input and output layer. Contrasting this with the work by Sommer et al. (2024), this suggests that MILE also exhibits most disconnected modes in the first and last layers. This analysis can help to assess whether sampling the multimodal posterior surface of BNNs is an infeasible problem due to the combinatorial explosion of modes with an increased depth of the network (which is not the case).

**Calibration** We also compute calibration errors (see Definition 2) and analyze coverage for credible intervals across various nominal coverage levels for the UCI benchmarks. Table 10 and Fig. 7 show that MILE achieves calibration quality comparable to the one of BDE, confirming its effectiveness in uncertainty quantification.

**Classification benchmark** For a comparison with BDE, we ran the smaller tabular classification tasks both using BDE and MILE. For the larger experiments (both considerably larger in dataset and model complexity), BDE would require weeks to run and is thus omitted. The results of the comparative study are given in Table 4 and suggest on-par performance of MILE with BDE and clearly superior performance to the DE baseline.

Table 4: Hold-out test performance of BDE, MILE and baselines on the two tabular classification tasks.

| Dataset | Accuracy (↑) | | | | | | LPPD (↑) | | | | | |
|---|---|---|---|---|---|---|---|---|---|---|---|---|
| | Avg. Single | | | Ensemble | | | Avg. Single | | | Ensemble | | |
| | DNN | Chain (BDE) | Chain (MILE) | DE | BDE | MILE | DNN | Chain (BDE) | Chain (MILE) | DE | BDE | MILE |
| Ionosphere | 0.930 | 0.955 | **0.958** | **0.958** | 0.958 | 0.958 | -0.404 | -0.172 | -0.168 | -0.309 | -0.172 | **-0.167** |
| Income | 0.843 | 0.850 | **0.851** | 0.846 | **0.851** | 0.851 | -0.334 | -0.315 | -0.315 | -0.318 | **-0.311** | -0.313 |

### A.1.2 ROBUSTNESS AND NUMERICAL STABILITY IN HIGH DIMENSIONS

Without our adjustments, MCLMC struggles with exploration and often fails to produce meaningful samples, especially in high-dimensional settings. To highlight this, we conduct an ablation study, assessing failure rates when applying MCLMC to BNNs without our proposed adjustments. Specifically, we use the same models as in Table 1 and run 100 chains each of MILE and naïve MCLMC on various datasets with differing parameter dimensions and with the same DE initialization, recording the percentage of chains that resulted in numerical issues (e.g., NaN values rendering all samples unusable). The results, given in Table 5, demonstrate the critical importance of our adjustments: MCLMC exhibits failure rates between 78% and 86%, while MILE consistently shows 0% failures across all datasets.

Table 5: Failure rates (NaN chains) for naïve MCLMC and MILE across different datasets.

| Dataset | MCLMC (% NaN Chains) | MILE (% NaN Chains) |
|---|---|---|
| Airfoil | 86% | **0%** |
| Concrete | 80% | **0%** |
| Energy | 78% | **0%** |
| Yacht | 85% | **0%** |

### A.1.3 FURTHER COMPARISONS

**Comparison with Path-Guided Particle-based Sampling**    We conduct an empirical comparison with the recently proposed Path-Guided Particle-based Sampling (PGPS, Fan et al., 2024). Following the experimental setup of Section 5.2.1 of Fan et al. (2024), we conduct BNN inference on 7 UCI classification datasets (Dua & Graff, 2017) and report the average negative log-likelihood (NLL) and accuracy. Table 6 contains the results that showcase a clear pattern. MILE performs at least as good in terms of accuracy as PGPS and is clearly superior in the NLL in most cases. Notably, running MILE for all experiments and replications takes less than 5 minutes on a consumer CPU, with many in under 1 minute. We compared PGPS and MILE in terms of runtime on the same hardware for the Sonar dataset across five independent runs as an example. While MILE achieved a runtime of $0.94\pm0.06$ minutes PGPS required $24.34\pm0.64$ minutes. The major factor for the runtime gap is the nested computation detailed in PGPS (Algorithm 3). For example, the authors chose 100k overall steps each with 100 optimization steps and 300 Langevin adjustments for the UCI benchmark. This incurs high computational costs, even without considering the additional overhead of the tuning of the PGPS hyperparameters $\alpha$ and $\beta$.

Table 6: Hold-out test performance of Path-Guided Particle-based Sampling and MILE on the UCI classification tasks of Table 1 and 4 of Fan et al. (2024) over five independent runs.

| Dataset | # Classes | # Rows | NLL ($\downarrow$) | | Accuracy ($\uparrow$) | |
|---|---|---|---|---|---|---|
| | | | **PGPS** | **MILE** | **PGPS** | **MILE** |
| SONAR | 2 | 207 | $\mathbf{0.536 \pm 0.014}$ | $0.979 \pm 0.094$ | $\mathbf{0.798 \pm 0.023}$ | $0.779 \pm 0.047$ |
| WINEWHITE | 7 | 4898 | $1.979 \pm 0.009$ | $\mathbf{1.110 \pm 0.014}$ | $0.452 \pm 0.010$ | $\mathbf{0.565 \pm 0.008}$ |
| WINERED | 6 | 1599 | $1.964 \pm 0.012$ | $\mathbf{1.060 \pm 0.037}$ | $0.594 \pm 0.018$ | $\mathbf{0.604 \pm 0.019}$ |
| AUSTRALIAN | 2 | 689 | $\mathbf{0.5042 \pm 0.013}$ | $0.486 \pm 0.087$ | $\mathbf{0.862 \pm 0.009}$ | $0.852 \pm 0.015$ |
| HEART | 5 | 302 | $0.943 \pm 0.030$ | $1.440 \pm 0.078$ | $0.256 \pm 0.142$ | $\mathbf{0.591 \pm 0.033}$ |
| GLASS | 6 | 213 | $1.685 \pm 0.030$ | $\mathbf{1.160 \pm 0.083}$ | $0.585 \pm 0.080$ | $\mathbf{0.643 \pm 0.063}$ |
| COVERTYPE | 7 | 8000 | $1.602 \pm 0.014$ | $\mathbf{0.717 \pm 0.024}$ | $0.590 \pm 0.095$ | $\mathbf{0.746 \pm 0.006}$ |

**Comparison with Symmetric Split HMC**    We also conduct an empirical comparison with Symmetric Split HMC (Sym-Split-HMC, Cobb & Jalaian, 2021). Symmetric Split HMC advances HMC but inherits the same hyperparameter sensitivity (e.g., depends on trajectory length and step size). These hyperparameters can limit the application in Bayesian neural network inference. In Cobb & Jalaian (2021), the authors use Bayesian Optimization (BO) to derive hyperparameters which introduces further complexity and a significant computational burden. Unlike MILE, Symmetric Split HMC employs an MH correction step, which further increases the computational costs. Another downside of Cobb & Jalaian (2021) is that with an increased number of batches, the computational requirements increase notably. Both approaches have merit, but their main goal and contribution differ considerably. Symmetric Split HMC focuses on memory scalability, while MILE optimizes speed and performance. Nevertheless, an empirical comparison is interesting.

We replicate the multi-class classification task for the Fashion-MNIST dataset from Table 2 with the CNN (v2) model using Sym-Split-HMC. We use the optimized hyperparameters reported in Cobb & Jalaian (2021), Section 5.3, for the same dataset and task. We conduct all experiments on the same hardware to ensure comparability of runtimes and report the results in Table 7. For a fixed amount of posterior samples, the performance of Symmetric Split HMC benefits from smaller batch sizes. However, as noted above and confirmed empirically, runtime increases notably for smaller batch sizes. We choose a batch size of 64 and run Symmetric Split HMC for 200 samples, requiring 15.5 hours. The intended goal of sampling 1000 samples (as with MILE) would take more than 3 days with this setting. For larger batches of 1024 images, we generate up to 3000 posterior samples for Symmetric Split HMC, but without a notable gain in performance. Regardless of the specification, it

becomes clear that MILE achieves considerably better performance in a fraction of the required time for Symmetric Split HMC (without considering the cost of running BO for hyperparameter tuning).

Table 7: Comparison of off-the-shelf MILE with Symmetric Split HMC on the Fashion-MNIST task using the CNNv2 model. Total Time does not consider the necessary BO step of Symmetric Split HMC.

| Method | Accuracy (↑) | LPPD (↑) | Post. Samples | Total Time |
|---|---|---|---|---|
| MILE | **0.925** | **−0.216** | 1000 | 1h 21min |
| Sym-Split-HMC (Batch size: 64) | 0.818 | −0.548 | 200 (+50 burn-in) | 15h 29min |
| Sym-Split-HMC (Batch size: 1024) | 0.820 | −0.525 | 1000 (+200 burn-in) | 7h 6min |
| Sym-Split-HMC (Batch size: 1024) | 0.813 | −0.513 | 3000 (+300 burn-in) | 18h 47min |

## A.2 EXPERIMENTAL SETUP AND FURTHER DETAILS

**Software** Our software is implemented in Python and mainly relies on the `jax` (Bradbury et al., 2018) and `BlackJAX` (Cabezas et al., 2024) libraries. We further use `Docker` for a reproducible experimental setup. Our code is available at `https://github.com/EmanuelSommer/MILE`.

**Compute environment** The experiments were run on two NVIDIA RTX A6000 GPUs and an AMD Ryzen™ Threadripper™ PRO 5000WX/3000WX CPU with 64 cores. Sampling 12 chains for most experiments allowed to parallelize the sampling on CPU such that multiple experiments can be run at the same time.

**Benchmark data** Table 8 gives an overview of the data, and associated tasks and provides all references.

Table 8: Overview of the used datasets with task description and references.

| ABBREV. | DATA SET | TASK | # OBS. | FEAT. | REFERENCE |
|---|---|---|---|---|---|
| A | AIRFOIL | REGRESSION | 1503 | 5 | DUA & GRAFF (2017) |
| B | BIKESHARING | REGRESSION | 17379 | 13 | FANAEE-T (2013) |
| C | CONCRETE | REGRESSION | 1030 | 8 | YEH (1998) |
| E | ENERGY | REGRESSION | 768 | 8 | TSANAS & XIFARA (2012) |
| P | PROTEIN | REGRESSION | 45730 | 9 | DUA & GRAFF (2017) |
| Y | YACHT | REGRESSION | 308 | 6 | ORTIGOSA ET AL. (2007); DUA & GRAFF (2017) |
| - | IONOSPHERE | BINARY-CLASS. | 351 | 34 | SIGILLITO ET AL. (1989) |
| - | INCOME | BINARY-CLASS. | 48842 | 14 | KOHAVI (1996) |
| - | IMDB | BINARY-CLASS. | 50000 | TEXT | MAAS ET AL. (2011) |
| - | MNIST | MULTI-CLASS. | 60000 | 28x28 | LECUN & CORTES (2010) |
| - | F(ASHION)-MNIST | MULTI-CLASS. | 60000 | 28x28 | XIAO ET AL. (2017) |

**Optimization & sampling** For all DE optimizations, we use ADAM with decoupled weight decay (Loshchilov & Hutter, 2019) and use the negative log-likelihood loss as objective. We employ early stopping on a validation set and use a 70% train, 10% validation and 20% test split if there is no predefined test set as for the MNIST and Fashion MNIST dataset. If not specified otherwise we use 12 DE members and 12 chains. For all NUTS-based experiments, we use a burn-in of 100 samples and collect 1000 posterior samples with a target acceptance rate of $0.8$. Also, we employ an isotropic standard Gaussian prior if not specified otherwise.

We do not adjust the effective number of samples in the MCLMC tuning even if we apply a considerable amount of thinning, i.e., for 10000 samples with a thinning interval of 10, resulting in 1000 final samples, we still use an ESS of 100. However, for less than 1000 final samples, we hold ESS fixed at 100 as a lower bound. We validated the robustness of this choice by various experiments and ablation studies discussed in Section 4.

**Regression tasks** We train distributional regression models for all regression tasks just as Izmailov et al. (2021); Sommer et al. (2024). That means, we parameterize the Gaussian likelihood with by the output neurons as location and log-scale. For the experiments aggregated in Table 3, Table 10

and Fig. 2, we use configurations as in Sommer et al. (2024), in particular, use a fully-connected neural network that has two hidden layers with 16 neurons each.

**Ablation studies** For the ablation studies, we use the larger UCI benchmark datasets `bikesharing` and `protein`. If not specified otherwise, we use slightly larger networks than before by considering three hidden layers of 16 neurons each. In order to analyze the behavior of the samplers, we further implement a slim and deeper network with 6 hidden layers of just 8 neurons each. The corresponding experimental results are reported in Fig. 6.

**Classification tasks** For the classification tasks, we follow the classical way of directly parameterizing the categorical distribution with as many output neurons as we have classes. For the tabular datasets `ionosphere` and `income`, we use a simple feed-forward neural networks with 2 (v1) and 4 (v2) hidden layers with 16 neurons each. We also consider new data modalities for the BDE, namely images and text. The corresponding architectures are described in Table 9. Moreover, we use 10 chains for CNNs, 8 chains for the sequential models v1-2, and 4 chains for the sequential model v3 for these larger networks. We save only 100 samples per chain to be more memory efficient and are thus able to showcase improvement even for a smaller overall amount of posterior samples. This is realized via thinning in both BDE and MILE.

**Convolutional neural networks** As convolutional neural network (CNN) architectures we choose a LeNet-5 (Lecun et al., 1998) architecture (CNNv2) and also consider a slightly smaller yet similar architecture (CNNv1). The architectures are described in detail in Table 9.

Table 9: CNN architectures.

|  | **CNNv2** | **CNNv1** |
|---|---|---|
| **Conv** | 6 filters, 5x5 kernel, padding 2, ReLU | 1 filter, 3x3 kernel, padding 2, ReLU |
| **Pooling** | 2x2 Avg Pooling, stride 2 | - |
| **Conv** | 16 filters, 5x5 kernel, no padding, ReLU | - |
| **Pooling** | 2x2 Avg Pooling, stride 2 | - |
| **FC** | 120 units, ReLU | 8 units, ReLU |
| **FC** | 84 units, ReLU | 8 units, ReLU |
| **FC** | Output units | 8 units, ReLU |
| **FC** | - | Output units |

**Sequential networks** Fig. 5 provides a schematic overview of the attention-based sequential model architecture. We explore two main configurations: one where all model parameters, including token and positional embeddings, are sampled (v1), and another using a fixed, pretrained embedding (v2,v3). Both models use a context length of 70 tokens, with padding or truncation for shorter or longer sequences. We trained a custom tokenizer with Byte-Pair Encoding (BPE, Gage, 1994), targeting vocabulary sizes of 1k and 10k tokens for v1 and v2-3, respectively. To balance model complexity, token embeddings were set to 48 dimensions for the fully sampled model v1 and 192 for the pretrained versions v2-3. Positional encodings are added before passing through an 8-head attention mechanism, with 64-dimensional query, key, and value vectors (Vaswani et al., 2017) for v1-2. For v3, we use a 10-head attention mechanism with 100-dimensional query, key and value vectors. After average pooling, a feed-forward network with one hidden layer (64 neurons for the full model v1, 32 for the pretrained version v2) or two hidden layers for v3 with 128 and 32 neurons output the logits.

**Prior induced regularization** As the prior acts as a regularizer during the sampling phase, we might exhibit performance degradation for larger classification models if the prior variance is chosen inappropriately small. For the larger CNN and ATT models, we therefore choose the standard isotropic Gaussians $\mathcal{N}(0, 0.1I)$ (CNNv2), $\mathcal{N}(0, 0.2I)$ (ATTv1,v2) and $\mathcal{N}(0, 0.4I)$ (ATTv3). While a dedicated study on the influence of priors within this framework is out of scope for this work, we think further tuning the prior variance or changing the prior distribution could be promising.

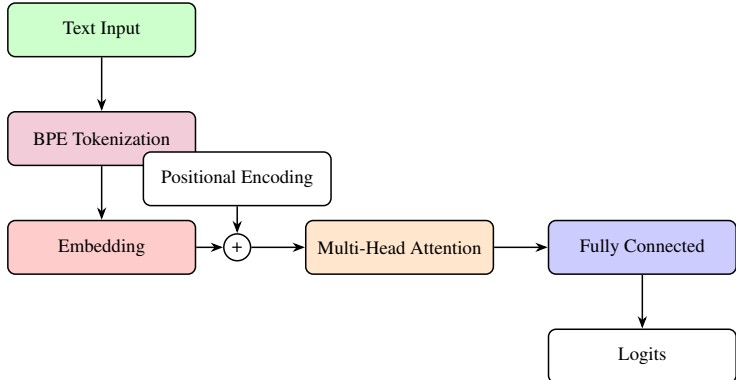

Figure 5: Schematic overview of the sequential attention-based model architecture (ATT) that is applied to the IMDB Dataset.

## A.3 DIAGNOSTICS

We report the BNN-specific diagnostics proposed in Sommer et al. (2024) in Fig. 6. The displayed chain variances are discussed in Appendix A.1.1.

**Effective sample size**  We observe very similar effective sample size (ESS) values for BDE as reported in Izmailov et al. (2021); Sommer et al. (2024). In most cases, MILE is on a same or slightly higher level of ESS. However, especially for weights that are close to the input and output, we observe a much higher ESS than for BDE. For the `airfoil` dataset, this ESS increase also given for deeper layers.

**Convergence of MILE and chainwise mixing**  Based on the results of Robnik et al. (2024), we know that MILE will provide the same convergence guarantees as long as the initialization is done randomly or its effect becomes negligible as $S \to \infty$ and the discretization error is MH-adjusted. As our work's focus is on empirical efficiency rather than guaranteed convergence, we a) do not use MH-adjustment but control discretization error using the EEVPD b) do not run chains for a large number of steps, and c) start chains using deep ensemble initializations. This is a compromise that will induce a bias in the sampling distribution but ensures a more stable behavior during sampling, and, in turn, increases the ESS.

To measure chainwise mixing, we proxy local chainwise convergence using the chainwise split metric $\widehat{cR}$ with a split factor of 4. Our results demonstrate that MILE clearly improves chainwise mixing. However, all values remain notably higher than the conventional cutoff thresholds of 1.1 and 1.01 (Vehtari et al., 2021).

## A.4 EVALUATION

**Predictive performance**  Following Gelman et al. (2014); Wiese et al. (2023) and Sommer et al. (2024), we choose the log posterior predictive density (LPPD) over a test set $\mathcal{D}_{\text{test}}$, defined as

$$\text{LPPD} = \frac{1}{n_{\text{test}}} \sum_{(\boldsymbol{y}^*, \boldsymbol{x}^*) \in \mathcal{D}_{\text{test}}} \log \left( \frac{1}{K \cdot S} \sum_{k=1}^{K} \sum_{s=1}^{S} p \left( \boldsymbol{y}^* | \boldsymbol{\theta}^{(k,s)}(\boldsymbol{x}^*) \right) \right) \tag{4}$$

in order to quantify the quality of the PPD approximation and UQ in general. Intuitively, the LPPD measures the average extent to which the predictive distribution accurately covers the observed labels.

Additionally, we use the root mean squared error (RMSE) for regression tasks and accuracy (ACC) for classification tasks to assess point predictions. While LPPD evaluates the overall fit of the predictive distribution, RMSE and ACC provide specific metrics for the accuracy of point predictions in their respective domains.

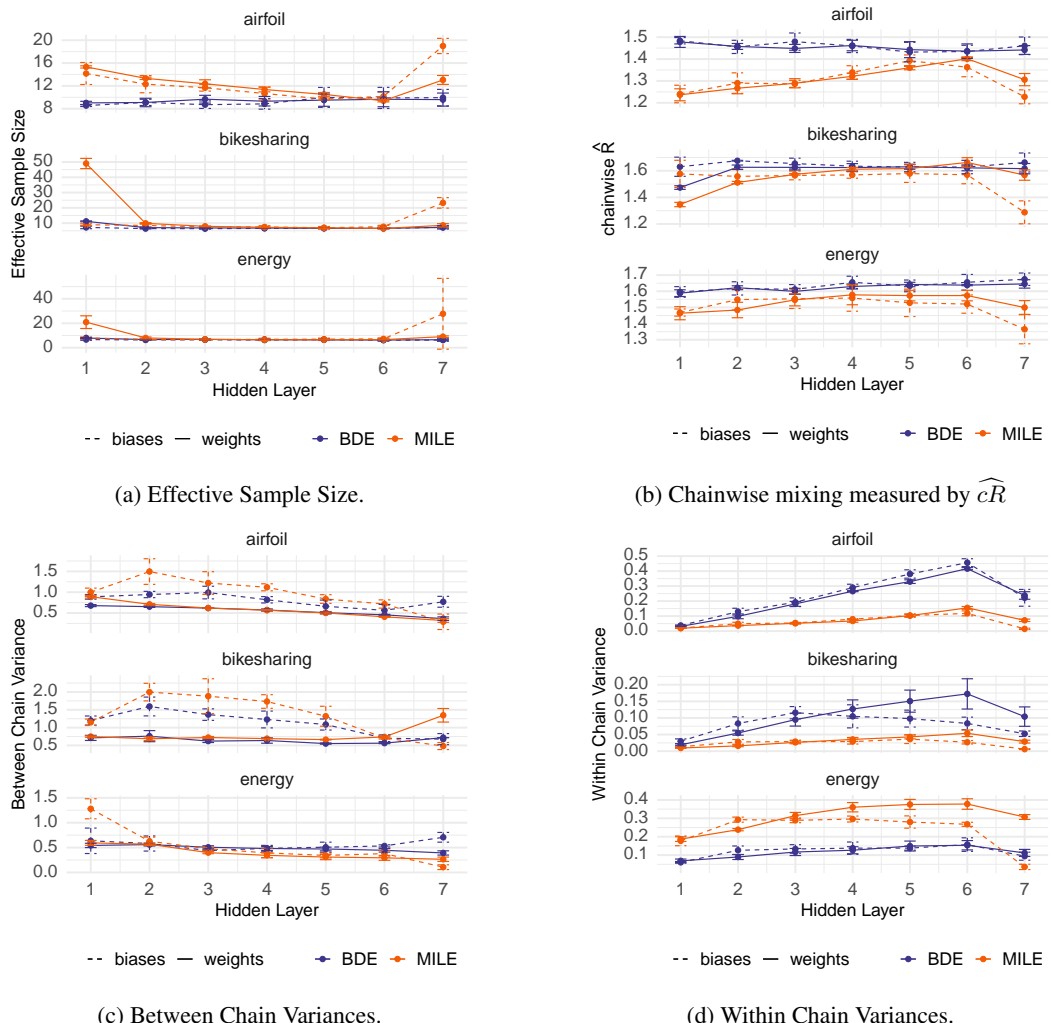

Figure 6: Different sampling diagnostics of a seven-layer BNN for three UCI benchmark datasets (in different rows) separated by layer (x-axis) over three data splits.

**Calibration**   Following Kuleshov et al. (2018), we define calibration and the empirical (squared) calibration error in the regression setting. Intuitively one expects samples from the true PPD to be contained in the Credibility Intervals (CIs) with the coverage probability of the CI. The following definition formalizes this.

**Definition 1 (Calibration)** *For some realized labeled dataset* $\mathcal{D} = \{(\boldsymbol{x}_i, y_i)\}_{i=1}^n \in \mathcal{X} \times \mathbb{R}$ *of random variables* $X, Y$, *we define a credible interval* $\mathcal{C}_{1-\alpha}(\boldsymbol{x}_*, \mathcal{D})$ *to be* ***calibrated*** *at level* $1 - \alpha \in (0, 1)$ *iff for* $y_* \sim p(\cdot \mid \boldsymbol{x}_*, \mathcal{D})$ *it holds that*

$$\mathbb{P}\big(y_* \in \mathcal{C}_{1-\alpha}(\boldsymbol{x}_*, \mathcal{D})\big) = 1 - \alpha. \tag{5}$$

*If* $y_* \in \mathcal{C}_{1-\alpha}(\boldsymbol{x}_*, \mathcal{D})$, *we say that the CI* $\mathcal{C}_{1-\alpha}(\boldsymbol{x}_*, \mathcal{D})$ *covers* $y_*$. *Thus calibrated models have correct* ***coverage*** *probabilities.*

This straightforwardly leads to the definition of the calibration error.

**Definition 2 (Calibration error)** *We define the empirical weighted* ***calibration error (CalE)*** *over the hold-out validation data set* $\mathcal{D}_*$ *as the root mean squared difference of nominal* $1 - \alpha_l$ *and*

Table 10: Mean Calibration Error for the DE baseline, BDE and MILE for six datasets. The nominal coverage levels used are 0.5, 0.75, 0.9 and 0.95. The experimental setup is identical to the one in Table 1.

| | Calibration Error ($\downarrow$) | | |
|---|---|---|---|
| **Dataset** | **DE** | **BDE** | **MILE** |
| Airfoil | $\mathbf{0.077 \pm 0.012}$ | $0.083 \pm 0.009$ | $0.086 \pm 0.018$ |
| Bikesharing | $0.125 \pm 0.005$ | $\mathbf{0.080 \pm 0.003}$ | $\mathbf{0.081 \pm 0.002}$ |
| Concrete | $0.066 \pm 0.002$ | $\mathbf{0.050 \pm 0.003}$ | $0.068 \pm 0.017$ |
| Energy | $0.215 \pm 0.015$ | $\mathbf{0.032 \pm 0.003}$ | $0.061 \pm 0.014$ |
| Protein | $\mathbf{0.054 \pm 0.011}$ | $\mathbf{0.056 \pm 0.002}$ | $\mathbf{0.057 \pm 0.003}$ |
| Yacht | $0.253 \pm 0.037$ | $\mathbf{0.188 \pm 0.032}$ | $\mathbf{0.133 \pm 0.059}$ |

*empirical $1 - \hat{\alpha}_l$ CI coverages over a range of $L$ relevant coverage levels $\alpha_1, \ldots, \alpha_L$:*

$$CalE(\mathcal{D}_*) = \left( \sum_{l=1}^{L} w_l \cdot (\hat{\alpha}_l - \alpha_l)^2 \right)^{\frac{1}{2}} \qquad (6)$$

$$\text{with } 1 - \hat{\alpha}_l = \frac{1}{|\mathcal{D}_*|} \sum_{(\boldsymbol{x}_*, y_*) \in \mathcal{D}_*} \mathbb{I}\{y_* \in \mathcal{C}_{1-\hat{\alpha}_l}(\boldsymbol{x}_*, \mathcal{D})\}, \qquad (7)$$

*where $w_l$ are normalized weights of the coverage levels which are commonly considered to be constant, i.e., $w_l = 1 \ \forall l \in [L]$.*

We report this calibration error in Table 10 for DEs, BDE and MILE for multiple datasets in the distributional regression setting. We consider the coverage levels 0.5, 0.75, 0.9, and 0.95. For most cases, we observe that the calibration error of MILE is on the same level as BDE and both methods often outperform the simple DE. For smaller datasets, however, estimating the empirical quantiles for small $\alpha$ is less robust due to limited test data size. Since the calibration error does not indicate whether the model is over- or underconfident, we also examine the coverage levels directly in Figure 7. The plots show a high variation in coverage quality of DE-based confidence intervals by exhibiting both strong structural under- and overconfidence, whereas the sampling-based methods are generally better calibrated. The larger datasets, `bikesharing` and `protein`, which more likely provide enough data for reliable empirical coverage estimates, are a good example of this: for `bikesharing`, DE is more underconfident, while for `protein`, it is overconfident in contrast to the sampling-based alternatives. A visual inspection reveals that both BDE and MILE tend to be slightly underconfident, which is however often preferred by the practitioners over structural overconfidence, as seen for example with DE in the `protein` dataset. All in all, a more careful analysis of calibration of MILE would be a great direction for future work.

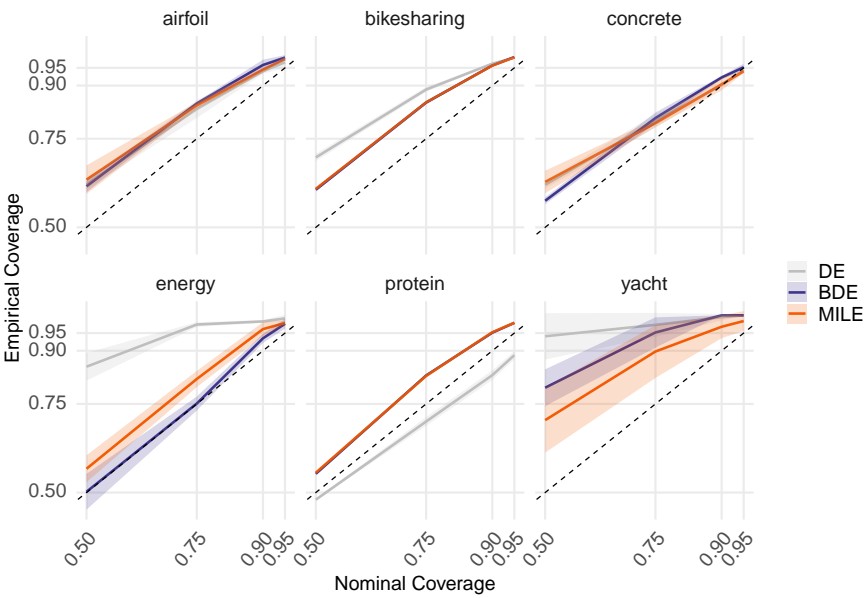

Figure 7: Mean and standard error of empirical coverage (y-axes) for the DE baseline, BDE and MILE for six datasets (facets). The nominal coverage levels used are 0.5, 0.75, 0.9 and 0.95 (x-axes). The experimental setup is identical to the one in Table 1.

