# OpenReview forum: "Microcanonical Langevin Ensembles: Advancing the Sampling of Bayesian Neural Networks"
_ICLR.cc/2025/Conference — ICLR 2025 Poster_

### Official Review · Reviewer_83Bk · 2024-10-30

**Soundness:** 3
**Presentation:** 3
**Contribution:** 3
**Rating:** 8
**Confidence:** 3

**Summary:**

The paper proposes a Microcanonical Langevin Ensemble (MILE) approach which adapts the MCLMC to high-dimensional posteriors common in modern Bayesian deep neural networks (BNN). The paper integrates optimization strategies from deep ensembles to carefully adjust components of MCLMC to make it scalable. Through extensive experiments, the paper shows the superiority of MILE in prediction and UQ on popular benchmarks.

**Strengths:**

I enjoyed reading the paper. Extensive experiments have been conducted where MILE has shown superior performance compared to its competitors. The main strengths of this paper are –
1.	This paper is a significant step towards scalable full Bayesian inference with deep neural networks.
2.	The paper explains the key contributions clearly.
3.	The paper is well written. The authors presented their approach clearly with an intuitive explanation of the key components.
4.	The authors presented extensive experiments to support their key claim which is the scalability of their approach for sampling high-dimensional Bayesian posteriors.

**Weaknesses:**

The paper can significantly improve if the authors can discuss/expand on the following points –
(1)	The paper discusses careful tunings of the components of MCLMC.  An ablation study is needed to understand which adjustments are more important than others. My guess is the main speed-up is due to the warm starting using the DE. However, this demands a proper study.
(2)	The work lacks theoretical justification. However, this is not a critical point for the paper as the authors have presented extensive experiments in support of their key claims. However, a discussion related to the convergence of the approach can improve the paper.
(3)	The numerical section seems to be lacking more competitive methods. “Path-Guided Particle-based Sampling” (ICML 24) can be another competitor approach that is proven to draw efficient samples from multi-modal Bayesian posteriors.

**Questions:**

In the “Performance results” paragraph the last sentence claims that “It is noteworthy that this is a big step for sampling-based inference, yielding a time complexity comparable to DE, while providing better and more principled uncertainty measures.” However, the time measured is on top of the DE fit as claimed in the caption of Table 1. Then the total time for MILE should be twice of DE. Please clarify.

---

> ### Author Response · Authors · 2024-11-16
> **Rebuttal by the Authors to the Question**
>
> Dear Reviewer 83Bk,
>
> Thank you for your positive feedback. We appreciate your thoughtful review and are pleased that you found the paper engaging and well-explained. Below, we address your questions and suggestions point by point.
>
> ### **Question on Time Complexity**
>
> > However, the time measured is on top of the DE fit as claimed in the caption of Table 1. Then the total time for MILE should be twice of DE. Please clarify
>
> The reviewer is correct that the sampling time needs to be added to the optimization time of DE, effectively doubling the DE time. We will make this more clear in an updated version of the manuscript. Note, however, that this is still the property we intended to highlight. Maintaining good sample quality while achieving sampling times that match optimization times is unprecedented for sampling-based inference in BNNs. Being able to sample at the same speed required by Adam-type optimization makes sampling strategies a viable alternative to optimization routines in deep learning applications and thereby creates an entirely new training paradigm of neural networks.

---

> ### Author Response · Authors · 2024-11-16
> **Rebuttal by the Authors to the Comments (1-2 of 3)**
>
> ### **Comments on Weaknesses**
>
> > (1) The paper discusses careful tunings of the components of MCLMC. An ablation study is needed to understand which adjustments are more important than others.
>
> We agree that examining the role of each adjustment is valuable:
>
> - **a) DE warmstart:** While the reviewer is correct that the DE startup notably improves the method, our reported speedups are a comparison between two methods that **both** use DE initialization. Hence, it is not correct to say that “the main speed-up is due to the warm starting using the DE” as both methods use warm starts via DEs.
> - **b) MCLMC without adjustment vs. with adjustment:** We did not provide such a study in the original submission as MCLMC without adjustment for BNNs will either be inefficient or in most cases not work at all. Thus, adjustments, such as the proposed tuning initialization, are integral to the numerical and overall feasibility of MILE in the BNN setting and cannot be entirely ablated.  We however agree that it would be essential to also show this in the paper. To this end, we have conducted a small benchmark study, sampling 100 chains of MILE and 100 naive MCLMC without adjustments for different datasets and dimensions of the parameter vector/model. We compare how many runs lead to numerical issues rendering all obtained samples useless (NaN).
>
> | **Dataset**  | **MCLMC (% NaN Chains)** | **MILE (% NaN Chains)** |
> |--------------|---------------------------|--------------------------|
> | **Airfoil**  | 86%                       | **0%**                   |
> | **Concrete** | 80%                       | **0%**                   |
> | **Energy**   | 78%                       | **0%**                   |
> | **Yacht**    | 85%                       | **0%**                   |
>
> - **c) Tuning parameters:** as discussed in 4.3, tuning parameters do not notably affect the procedure once the adjustments from b) are incorporated
>
> > (2) The work lacks theoretical justification. However, this is not a critical point for the paper as the authors have presented extensive experiments in support of their key claims. However, a discussion related to the convergence of the approach can improve the paper.
>
> We thank the reviewer for pointing this out. While the focus of our work is on empirical efficiency rather than guaranteed convergence, we acknowledge the importance of discussing MILE’s convergence properties. To check convergence, we examined local mixing using split-chain Rhat (see, e.g., Appendix A.3), which indicates improved local convergence over competitor methods. Apart from expanding Appendix A.3 to better convey our convergence diagnostics and rationale, we will further add a discussion on the theoretical convergence of our method. While the reviewer is correct that we do not provide any theoretical guarantees at this point, a theoretical argument would “only” require the analysis of the effect of initialization as, e.g., done in Wild et al. (2024) for deep ensembles.

---

> ### Author Response · Authors · 2024-11-16
> **Rebuttal by the Authors to the Comments (3rd of 3)**
>
> > (3) The numerical section seems to be lacking more competitive methods. “Path-Guided Particle-based Sampling” (ICML 24) can be another competitor approach that is proven to draw efficient samples from multi-modal Bayesian posteriors.
>
> Including “Path-Guided Particle-based Sampling” (PGPS) as a competitor method is an excellent suggestion. We will discuss PGPS in the related work section together with another related paper from ICML on Liouville Flow Importance Sampler (Tian et al., 2024). We also conducted an empirical comparison, using the experimental setup from Section 5.2.1 of their paper, which we will add to our manuscript.  We performed BNN inference on seven UCI classification datasets, evaluating average negative log-likelihood (NLL) and accuracy. Notably, MILE completed all experiments and replications in under 5 minutes on a small PC CPU, with many tasks taking less than 1 minute. The results, presented in the table below, demonstrate that MILE achieves comparable or better accuracy while consistently outperforming PGPS in NLL across most cases, thus, again highlighting its competitive performance.
>
> | **Dataset**      | **NLL (↓)** PGPS         | **NLL (↓)** MILE         | **Accuracy (↑)** PGPS       | **Accuracy (↑)** MILE       |
> |-------------------|--------------------------|--------------------------|-----------------------------|-----------------------------|
> | **SONAR**        | **0.536 ± 0.014**        | 0.979 ± 0.094           | **0.798 ± 0.023**           | **0.779 ± 0.047**           |
> | **WINEWHITE**     | 1.979 ± 0.009           | **1.110 ± 0.014**        | 0.452 ± 0.010               | **0.565 ± 0.008**           |
> | **WINERED**       | 1.964 ± 0.012           | **1.060 ± 0.037**        | **0.594 ± 0.018**           | **0.604 ± 0.019**           |
> | **AUSTRALIAN**    | **0.504 ± 0.013**        | **0.486 ± 0.087**        | **0.862 ± 0.009**           | **0.852 ± 0.015**           |
> | **HEART**         | 0.943 ± 0.030           | **1.440 ± 0.078**        | 0.256 ± 0.142               | **0.591 ± 0.033**           |
> | **GLASS**         | 1.685 ± 0.030           | **1.160 ± 0.083**        | **0.585 ± 0.080**           | **0.643 ± 0.063**           |
> | **COVERTYPE**     | 1.602 ± 0.014           | **0.717 ± 0.024**        | 0.590 ± 0.095               | **0.746 ± 0.006**           |

---

> ### Author Response · Authors · 2024-11-16
> **Rebuttal Summary and References**
>
> We thank the reviewer again for the great suggestions and comments. We think these helped us to further improve our manuscript. Please let us know if our response addresses your questions and concerns, and if there is anything else we need to clarify.
>
> **References:**
>
> Fan et al., 2024: Path-Guided Particle-based Sampling, ICML 2024
>
> Tian et al., 2024: Liouville Flow Importance Sampler, ICML 2024
>
> Wild et al., 2024: A rigorous link between deep ensembles and (variational) Bayesian methods, NeurIPS 2024

---

> ### Comment · Reviewer_83Bk · 2024-11-20
> **Appreciate the comparison with PGPS**
>
> I appreciate the author's thorough comparison with the PGPS method and clarifying my other questions. Indeed MILE shows competitive performance vs PGPS (in most cases MILE wins). The very small runtime for MILE is also very interesting and supports the main contribution of the paper. Can the authors specify the hyperparameters used for PGPS? Also, please comment on the efficiency of MILE vs PGPS.
>
> Since this was one of my main concerns for the paper, I will increase the score.

---

> > ### Author Response · Authors · 2024-11-21
> > **Appreciate Your Feedback and PGPS Efficiency Discussion**
> >
> > Dear Reviewer 83Bk,
> >
> > We sincerely appreciate your thoughtful comments and your recognition of the contributions of our work, as well as your decision to raise the score. Below, we address your question regarding the hyperparameters used for PGPS and the efficiency comparison with MILE.
> >
> > **Hyperparameters for PGPS**
> >
> > The PGPS paper does not explicitly report the values of the hyperparameters $\alpha$ and $\beta$ used in their UCI experiments, nor are these details provided in their GitHub repository. To ensure a fair and meaningful comparison, we followed their reported data preprocessing steps (well-documented in their code repository) for the application of MILE. This allows us to directly compare against the reported performance metrics from Tables 1 and 4 in their paper.
> >
> > **Efficiency Comparison**
> >
> > Furthermore, in order to compare the runtime we ran PGPS on the same hardware used for MILE with $\alpha = 0.2$ and $\beta = 0.5$. Other hyperparameters, such as the number of steps for optimization and Langevin adjustments, were kept as provided in their code base and were not modified (specified in their code here: https://github.com/MingzhouFan97/PGPS/blob/main/experiments/UCI/inference_pggf.py#L84).
> >
> > We compared PGPS and MILE in terms of runtime for the Sonar dataset across five independent runs as an example. The results are as follows:
> >
> > - **MILE**: 0.94 ± 0.06 minutes
> > - **PGPS**: 24.34 ± 0.64 minutes
> >
> > The major factor for the runtime gap is the nested computation detailed in PGPS (Algorithm 3). For example, the authors chose 100k overall steps each with 100 optimization steps and 300 Langevin adjustments for the UCI benchmark. This incurs high computational costs, even without considering the additional overhead of hyperparameter tuning.
> >
> > An additional factor is that MILE is implemented in JAX, which takes full advantage of modern hardware acceleration. In contrast, PGPS is implemented in PyTorch, which may introduce inefficiencies when scaling to large, nested computations.
> >
> > We believe this comparison underscores the efficiency advantage of MILE. By design, MILE avoids such nested computations and relies on a streamlined and parallel approach. Importantly, MILE does not only yield faster but also competitive or better performance as demonstrated by the benchmark results.
> >
> > **Summary**
> >
> > We hope this addresses your questions. Please let us know if there are any additional aspects we can clarify. Thank you once again for your supportive and constructive review!

---

### Official Review · Reviewer_k13v · 2024-11-02

**Soundness:** 3
**Presentation:** 3
**Contribution:** 2
**Rating:** 6
**Confidence:** 2

**Summary:**

This paper adapts the MCLMC algorithm to sample from BNN posteriors. The authors propose a series of changes to the 3-stage tuning scheme in the original paper of MCLMC algorithm and name the adapted algorithm MILE. The authors show that by doing these, MILE demonstrates superior predictive performance, improved uncertainty quantification and improved runtime.

**Strengths:**

1. The paper is clear and detailed in terms of related work and the method.
2. The empirical results look comprehensive and solid.

**Weaknesses:**

1. The authors claim that their method is “tuning-free”, but in 3.2, several parameters are still tuned. Maybe I misunderstood what the authors mean by “tuning-free”

2. I feel that the author could elaborate more on which of the benefits of MILE mentioned in section 4 is inherited from the MCLMC algorithm, and which of them result from the authors’ adaptation.

Minor:
1. The acronym ESS first appear in 3.2.3 without any explanation

**Questions:**

1. What is d in $(d-1)^{-1}$ in Eq. 3?

---

> ### Author Response · Authors · 2024-11-16
> **Rebuttal by the Authors**
>
> Dear Reviewer k13v,
>
> Thank you for your positive feedback. Below, we respond to your comments and questions one by one.
>
> ### **Questions**
>
> > What is d in (d−1)−1 in Eq. 3?
>
> The d in Eq. 3 refers to the dimensionality of the parameter vector θ, as noted at the start of Section 2. We will better clarify this in the text.
>
> ### **Comments on Weaknesses**
>
> > The authors claim that their method is “tuning-free”, but in 3.2, several parameters are still tuned.
>
> **"Tuning-Free" Terminology:** While our method does involve a tuning phase, it is designed to be automated so that no manual tuning by the practitioner is required. This allows MILE to function as a reliable, **off-the-shelf** learning approach. We will try to highlight this again in the manuscript to avoid misunderstandings. Thank you for bringing this up.
>
> > I feel that the author could elaborate more on which of the benefits of MILE mentioned in section 4 is inherited from the MCLMC algorithm, and which of them result from the authors’ adaptation.
>
> We adapted MCLMC on multiple fronts.
>
> - **Multimodality**: Without the DE initialization, MCLMC will not be able to explore the posterior of BNNs sufficiently fast.
> - **Robustness, numerical stability** in high dimensions: Without our adjustments, MCLMC will not only suffer from insufficient exploration but fail to produce meaningful samples. To emphasize this, we ran an ablation study requested by Reviewer 83Bk, showing how often MCLMC will fail in an application to BNNs without our adjustment. For this, we sample 100 chains of MILE and 100 naive MCLMC without adjustments for different datasets and dimensions of the parameter vector/model. We compare how many runs lead to numerical issues rendering all obtained samples useless (NaN).
>
> | **Dataset**  | **MCLMC (% NaN Chains)** | **MILE (% NaN Chains)** |
> |--------------|---------------------------|--------------------------|
> | **Airfoil**  | 86%                       | **0%**                   |
> | **Concrete** | 80%                       | **0%**                   |
> | **Energy**   | 78%                       | **0%**                   |
> | **Yacht**    | 85%                       | **0%**                   |
>
> - **Enhanced Scalability**: We improve scalability by addressing critical bottlenecks, such as the FFT component within the tuning phase, further strengthening MILE's suitability for practical BNN applications.
> - **MILE’s speed and performance**: The speed at which MILE achieves state-of-the-art performance on various BNN inference tasks is unprecedented for full sampling-based inference and can be considered a novelty by itself. This not only becomes clear from the comparison with BDE in the paper but also in the newly added comparison with Split HMC provided to Reviewer e8p6.
>
> ### **Minor comments**
>
> **ESS acronym not defined:** Thank you for pointing this out. We will define ESS (Effective Sample Size) when it first appears in Section 3.2.3.
>
> ---
>
> Thank you for carefully reading our manuscript and helping us to refine our explanations and terminology in the manuscript. Please let us know if our response addresses your questions and concerns, and if there is anything else we need to clarify.

---

> > ### Author Response · Authors · 2024-11-25
> >
> > Dear Reviewer k13v,
> >
> > Thank you again for providing highly valuable feedback on our manuscript. We are pleased that you find our manuscript clear and detailed and acknowledge our empirical results.
> >
> > As the discussion period will end in approximately ~48 hours, we would like to ask if our answers so far have successfully addressed your concerns and answered your questions.
> >
> > In summary, we addressed the points raised by you as follows:
> >
> > - **Dimensionality in Eq. 3**: Clarified that $d$ refers to the dimensionality of the parameter vector $\theta$.
> > - **"Tuning-Free" Terminology**: Explained that "tuning-free" refers to automated parameter tuning without practitioner intervention, highlighting MILE’s off-the-shelf reliability.
> > - **Adaptations of MCLMC**: Detailed how MILE’s benefits (e.g., multimodality, robustness, numerical stability, scalability) stem from critical adaptations, supported by an ablation study showing MCLMC's failure rates versus MILE's robustness.
> > - **Enhanced Scalability and Speed**: Showcased improvements in scalability and highlighted MILE’s unprecedented speed and state-of-the-art performance in sampling-based BNN inference.
> > - **ESS Acronym**: Noted that ESS (Effective Sample Size) will be defined clearly in Section 3.2.3.
> >
> > If there are any remaining concerns, please let us know.

---

> > > ### Comment · Reviewer_k13v · 2024-11-26
> > >
> > > Thank you for the reply. I read through the authors' conversation with all other reviewers. I think there are still concerns that are not resolved, so I will remain my rating.

---

### Official Review · Reviewer_qCdd · 2024-11-04

**Soundness:** 2
**Presentation:** 3
**Contribution:** 2
**Rating:** 6
**Confidence:** 3

**Summary:**

The authors propose an adaptation of the Microcanonical Langevin Monte Carlo method for Bayesian Neural Networks.

**Strengths:**

Clarity:
The paper is clearly written and well-organized.

Quality:
The evaluation is conducted meticulously, with numerous ablations considered. The results effectively justify the proposed method.

**Weaknesses:**

Novelty:
The paper appears to be an incremental modification of the Microcanonical Langevin Monte Carlo method.

Significance:
Proposing efficient sampling methods for Bayesian Neural Networks is an important problem. However, the main results of the paper are achieved through careful parameter settings, particularly in establishing ensemble methods, tuning size, step size, energy variance scheduler, sample size, and so on. Each of these parameters is known to be crucial for obtaining better results. This makes the paper seem somewhat ad-hoc to me, lacking sufficient significance for developing improved BNN samplers.

**Questions:**

The authors proposed a set of techniques to adapt the sampler for Bayesian Neural Network solvers. Is my understanding correct that to adapt Microcanonical Langevin Monte Carlo to BNNs, all we need to do is configure the parameters of the MCLMC and treat it as an ensemble method to reduce initialization error?

---

> ### Author Response · Authors · 2024-11-16
> **Rebuttal by the Authors**
>
> Dear Reviewer qCdd,
>
> Thank you for your feedback. Below, we respond to your comments on the novelty and significance of our approach.
>
> ### **Weaknesses and Questions**
>
> > The paper appears to be an incremental modification of the Microcanonical Langevin Monte Carlo method.
>
> We thank the reviewer for this thoughtful comment but politely disagree. We adapted MCLMC on multiple fronts.
>
> - **Multimodality**: Without the DE initialization, MCLMC will not be able to explore the posterior of BNNs sufficiently fast.
> - **Robustness, numerical stability** in high dimensions: Without our adjustments, MCLMC will not only suffer from insufficient exploration but fail to produce meaningful samples. To emphasize this, we ran an ablation study requested by Reviewer 83Bk, showing how often MCLMC will fail in an application to BNNs without our adjustment. For this, we sample 100 chains of MILE and 100 naive MCLMC without adjustments for different datasets and dimensions of the parameter vector/model. We compare how many runs lead to numerical issues rendering all obtained samples useless (NaN).
>
> | **Dataset**  | **MCLMC (% NaN Chains)** | **MILE (% NaN Chains)** |
> |--------------|---------------------------|--------------------------|
> | **Airfoil**  | 86%                       | **0%**                   |
> | **Concrete** | 80%                       | **0%**                   |
> | **Energy**   | 78%                       | **0%**                   |
> | **Yacht**    | 85%                       | **0%**                   |
>
> - **Enhanced Scalability**: We improve scalability by addressing critical bottlenecks, such as the FFT component within the tuning phase, further strengthening MILE's suitability for practical BNN applications.
> - **MILE’s speed and performance**: The speed at which MILE achieves state-of-the-art performance on various BNN inference tasks is unprecedented for full sampling-based inference and can be considered a novelty by itself. This not only becomes clear from the comparison with BDE in the paper but also in the newly added comparison with Split HMC provided to Reviewer e8p6.
>
> > Proposing efficient sampling methods for Bayesian Neural Networks is an important problem. However, the main results of the paper are achieved through careful parameter settings [...] This makes the paper seem somewhat ad-hoc to me
>
> MILE is designed as a **reliable off-the-shelf option**, similar to NUTS and Adam, offering high performance with minimal tuning required for reliable, scalable BNN inference. While we use the approach of Sommer et al. (2024) to induce enough exploration in the sampling procedure, other warm start options are potentially possible. If these options are equally or less computationally costly than DE starting values, MILE enables good sample quality for highly multimodal posteriors while achieving sampling times that match optimization times. This is unprecedented for sampling-based inference in BNNs. Being able to sample at the same speed required by Adam-type optimization makes sampling strategies a viable alternative to optimization routines in deep learning applications and thereby creates an entirely new training paradigm of neural networks. So to answer the reviewer’s question: the ingredients are MCLMC and ensembles similar to Adam being a combination of momentum and RMSprop. However, it takes considerable effort to adapt and combine these methods in order to perform on a scale that yields overhead the community is willing to accept when optimizing (sampling) neural networks. We strongly believe that the two components to achieve this are MCLMC and an ensemble of diverse starting values.
>
> ---
>
> Thank you again for your constructive feedback. Please let us know if our response addresses your questions and concerns, and if there is anything else we need to clarify.

---

> > ### Comment · Reviewer_qCdd · 2024-11-24
> > **Response to official comment by authors**
> >
> > I appreciate the authors response. However, my main concerns about the novelty of the work remain unaddressed. The question in my review was ignored. In the response, the authors mention some connection to the Adam optimizer. However, simply stating that your ingredients are similar to Adam does not demonstrate the novelty of your method; rather, it suggests that a set of well-known techniques was applied. Additionally, the authors of the Adam paper provided comprehensive theoretical justification for their modifications (see Corollary 4.2 and the appendix [1]). In contrast, no theoretical justification is provided for your results. I appreciate the authors' efforts to conduct additional experiments within a short period, but no details about these experiments are provided. While the main contribution of the method is practical, no source code, not even for the toy experiments, is included. Therefore, I will maintain my score.
> >
> > [1] Adam: A method for stochastic optimization. https://arxiv.org/pdf/1412.6980

---

> > > ### Author Response · Authors · 2024-11-25
> > >
> > > Dear reviewer qCdd,
> > >
> > > Thank you for taking the time to read our response and provide further feedback. Please find our detailed response below:
> > >
> > > >  simply stating that your ingredients are similar to Adam does not demonstrate the novelty of your method
> > >
> > > We apologize if our response was unclear. We are NOT claiming that our ingredients are similar to Adam. Instead, our point was that, like many established methods, MILE builds upon existing techniques while requiring significant effort to adapt and integrate these components effectively. For example, our additional benchmark demonstrates that naive ensembling of MCLMC without our adaptations does not produce meaningful results.
> > >
> > > > My main concerns about the novelty of the work remain unaddressed. The question in my review was ignored.
> > >
> > > We thank the reviewer for this comment but politely disagree that your question was ignored. Below, we try to reformulate our response to make the novelty more clear:
> > >
> > > - We found that MCLMC does not work for highly multimodal problems found for BNNs and previous HMC-based approaches as proposed in Sommer et al. (ICML 2024) are slow and exhibit unpredictable runtimes. Subsequently, we adapted MCLMC as a promising alternative to state-of-the-art NUTS.
> > > - These adaptions involve a) combining MCLMC with an ensemble approach, but also changing the originally proposed tuning phase, by modifying b) the step size, c) the energy variance scheduler, d) the effective sample size;
> > > - furthermore, we e) eliminate a bottleneck in the originally proposed Phase III by replacing the FFT with a more efficient version.
> > > - **Our main contribution is thus**
> > >     - a reliable, off-the-shelf approach for multimodal problems (see also new results provided in our previous response)
> > >     - for fast, tuning-free, and accurate sampling-based inference for BNNs (empirical evidence provided in the original submission and replies to Reviewers e8p6 and 83Bk).
> > >
> > >
> > > > In contrast, no theoretical justification is provided for your results.
> > >
> > > Thank you for raising this concern. While our work focuses primarily on empirical efficiency, we acknowledge the importance of discussing theoretical aspects. In response to comments from Reviewer 83Bk, we expanded Section A.3 to address MILE’s convergence properties based on algorithmic choices discussed in detail in Sections 2 and 3.
> > >
> > > > but no details about these experiments are provided. While the main contribution of the method is practical, no source code, not even for the toy experiments, is included.
> > >
> > > We believe this is a misunderstanding. Section A.2 of the manuscript includes a link to our anonymized repository (https://anonymous.4open.science/r/MILE-1CC1/README.md), which contains the source code for all experiments, including the additional ones conducted during the rebuttal phase. We will ensure this link is made more prominent in future revisions.
> > >
> > > —
> > >
> > > Thank you again for your feedback. Please let us know if this response addresses your concerns or if there are additional aspects you would like us to clarify.
> > >
> > > Best regards,
> > > The Authors

---

### Official Review · Reviewer_e8p6 · 2024-11-04

**Soundness:** 3
**Presentation:** 3
**Contribution:** 3
**Rating:** 3
**Confidence:** 4

**Summary:**

This interesting paper considers the application of microcanonical Langevin Monte Carlo, a variant of HMC, to sample from Bayesian neural network posteriors. The main idea of microcanonical LMC is that the velocity's norm is fixed and not changing unlike in HMC, which is supposedly helping with more stable exploration and allows for larger step sizes in the presence of steep landscapes. The original algorithm was proposed in the 2023 paper "Fluctuation without dissipation: Microcanonical Langevin Monte Carlo". The main contribution here is to consider an ensembled variant of that algorithm, and extensive numerical experiments on various Bayesian neural networks to show the practical performance of this method.

**Strengths:**

Microcanonical Langevin Monte Carlo is an interesting idea, and the numerical performance shows good improvements over NUTS, with impressive results in terms of predictive log posterior values.

**Weaknesses:**

The only novelty in terms of algorithmic development is the use of ensembling, Microcanical LMC was already proposed in the previous paper "Fluctuation without dissipation: Microcanonical Langevin Monte Carlo".

The method is not using minibatches, but instead, each step needs a full gradient, meaning that the cost of Bayesian neural networks is at least 2-3 orders of magnitude higher than using deterministic neural networks. Hence the practicality of the algorithm at this point is questionable.

There was no comparison done with the earlier method "Scaling Hamiltonian Monte Carlo inference for Bayesian neural networks with symmetric splitting" by Cobb et al., who managed to get HMC working with no bias using minibatches at each step, and an accept/reject step. It would be interesting to explore whether such a variant could be extended to Microcanonical LMC.

A major problem is that the implemented algorithm is not clearly described in the paper, and it's not clear whether a Metropolis/Hastings step is used or not, but the authors did not claim they used one so I presume it's not used. The previous paper "Fluctuation without dissipation: Microcanonical Langevin Monte Carlo", (https://arxiv.org/pdf/2303.18221) claims that their Euler-Mayurama discretization (15) exactly preserves the invariant distribution, so there is no need for accept/reject step. The precise form of the implementation of (15) is not stated there either. I am very skeptical that an explicit discretization only using a single gradient evaluation per iteration can exactly preserve the invariant distribution, hence I am on the opinion that this is not a fully explicit scheme, unless the authors convince me otherwise in the rebuttal.

**Questions:**

State the implemented algorithm precisely, including how many gradient evaluations are used per step, and state whether it preserves the invariant distribution. State whether accept/reject step is needed or not.

How does the method compares with "Scaling Hamiltonian Monte Carlo inference for Bayesian neural networks with symmetric splitting"?

---

> ### Author Response · Authors · 2024-11-16
> **Rebuttal by the Authors to the Questions**
>
> Dear Reviewer e8p6,
>
> Thank you for your thorough and thoughtful review. Below, we address each of your comments and questions in detail.
>
> ### **Algorithm Description and Dynamics**
>
> > State the implemented algorithm precisely, including how many gradient evaluations are used per step, and state whether it preserves the invariant distribution. State whether accept/reject step is needed or not.
>
> 1. **Algorithm Implementation:** We follow the dynamics proposed by [1] and adapt the MCLMC implementation from [2]. For each sample, our implementation requires two gradient evaluations as we use the minimal norm integrator.
> 2. **Invariant Distribution:** Our method is not a fully exact scheme. This is an intentional design choice, leveraging the resulting efficiency gains. We will further clarify this point in the manuscript.
> 3. **Accept/Reject Step:** We adopt an unadjusted sampling approach without MH correction, prioritizing computational efficiency over exact discretization error correction. This is discussed in the background section of the paper. We argue that the discretization error targeted by the MH correction is comparatively small compared to the initialization and Monte Carlo error. Hence, there is little benefit of using a correction in light of its computational cost. We will revise the manuscript to better highlight the rationale behind our approach.
>
> ### **Comparison with Cobb et al. (2021) [3]**
>
> > How does the method compares with "Scaling Hamiltonian Monte Carlo inference for Bayesian neural networks with symmetric splitting"?
>
> We thank the reviewer for the additional reference. We will include the below discussion in a revised version of the manuscript:
>
> 1. **Limitations of Symmetric Split HMC:** Symmetric Split HMC advances HMC, but inherits the same hyperparameter sensitivity (e.g., depends on trajectory length and step size). As discussed in detail in [4], these hyperparameters can limit the application in Bayesian neural network inference. In [3], the authors use Bayesian Optimization to derive hyperparameters which introduces further complexity and a significant computational burden. Unlike MILE, Symmetric Split HMC employs an MH correction step, which further increases the computational costs. Another downside of [3] is that with an increased number of batches, the computational requirements increase drastically. Both approaches have merit, but their main goal and contribution differ considerably.  Symmetric Split HMC focuses on memory scalability, while MILE optimizes speed and performance.
> 2. **Empirical Comparison:** We have conducted a benchmark with Symmetric Split HMC (using the official hamiltorch implementation) on our CNN (v2) architecture on Fashion-MNIST using the optimized parameters for the same task from Section 5.3 of [3]. All models are run on the same hardware to ensure runtime comparability. The results from this benchmark are displayed in the table below. We found that for a fixed amount of posterior samples, Symmetric Split HMC performs better for smaller batch sizes. However, as noted earlier and confirmed empirically, runtime increases dramatically for smaller batch sizes. We chose a batch size of 64 and ran Symmetric Split HMC for 200 samples. This required 15.5 hours, after which we had to stop the routine. The intended goal of sampling 1000 samples (as in our paper) would take more than 3 days. For larger batch sizes, we were able to generate 1000 posterior samples for Symmetric Split HMC, but without a notable gain in performance. Regardless of the specification, it becomes clear that MILE achieves considerably better performance in a fraction of the required time for Symmetric Split HMC, without even considering the cost of the Bayesian Optimization necessary for the practical application of Symmetric Split HMC.
>
> | **Method**                            | **Accuracy (↑)** | **LPPD (↑)**   | **Post. Samples**       | **Total Time** |
> |---------------------------------------|------------------|----------------|--------------------------|----------------|
> | **MILE**                              | **0.925**        | **-0.216**     | 1000                    | 1h 21min       |
> | **Symmetric Split HMC (Batch size: 64)**   | 0.818            | -0.548         | 200 (+50 burn-in)       | 15h 29min      |
> | **Symmetric Split HMC (Batch size: 1024)** | 0.820            | -0.525         | 1000 (+200 burn-in)     | 7h 6min        |

---

> > ### Comment · Reviewer_e8p6 · 2024-11-23
> > **Response to rebuttal**
> >
> > Dear Authors,
> >
> > Thank you for your efforts to answer my questions related to the paper.
> >
> > Unfortunately, your answers do not state precisely what is the algorithm you have implemented (MILE).
> > You state an SDE, and it is not explained anywhere in the paper how is this discretized.
> > You say that you are following the same dynamics as in the paper "Fluctuation without dissipation: Microcanonical Langevin
> > Monte Carlo" by Robnik and Seljak.
> >
> > It could be that you did not notice this, but that paper is full of mistakes, such as the claim of Theorem 3 that the Euler-Mayurama discretization exactly preserves the invariant distribution is completely wrong. The code for the paper of Robnik and Seljak attempts to solve the SDE accurately by using an adaptive step size high order Runge-Kutte method, which takes multiple gradient evaluations per step. There is no mention of this in their paper.
> >
> > I strongly feel that the precise statement of the discretization of the SDE should be included in the paper, otherwise it is absolutely unclear what is implemented in the code. You cannot expect readers to look at thousands of lines of code to figure out what is the key algorithm of the paper actually doing.
> >
> > It is unfortunate that you have not been more forthcoming with the precise statement of the discretization, and did not do any experiments to evaluate the bias of the method. It is commendable that you attempted to compare the method with Symmetric Split HMC, but I am skeptical about the fact that the differences between the performance in terms of accuracy are so significant, as the Symmetric Split HMC has no bias (due to the Metropolis/Hastings step). With better tuning of the step size and the number of steps, as well as the batch size, you should be able to obtain similar accuracy for all unbiased methods, but perhaps the efficiency in terms of number of effective samples per second could be significantly better for some methods.
> >
> > For these reasons, I will keep my mark.

---

> ### Author Response · Authors · 2024-11-16
> **Rebuttal by the Authors to Comments, Rebuttal Summary and References**
>
> ### **Additional Comments**
>
> >  only novelty in terms of algorithmic development is the use of ensembling
>
> **Novelty Beyond Ensembling:** While ensembling is a key component, our method also incorporates several essential modifications of MCLMC and algorithmic adjustments that enable the sampler to be applied to BNNs in a numerically stable, more scalable, and efficient way. To emphasize this, we ran an ablation study requested by Reviewer 83Bk, showing how often MCLMC will fail in an application to BNNs without our adjustment. For this, we sample 100 chains of MILE and 100 naive MCLMC without adjustments for different datasets and dimensions of the parameter vector/model. We compare how many runs lead to numerical issues rendering all obtained samples useless (NaN).
>
> | **Dataset**  | **MCLMC (% NaN Chains)** | **MILE (% NaN Chains)** |
> |--------------|---------------------------|--------------------------|
> | **Airfoil**  | 86%                       | **0%**                   |
> | **Concrete** | 80%                       | **0%**                   |
> | **Energy**   | 78%                       | **0%**                   |
> | **Yacht**    | 85%                       | **0%**                   |
>
> Furthermore, as also shown in the comparison with Symmetric Split HMC, the speed at which MILE achieves state-of-the-art performance on various BNN inference tasks is unprecedented for full sampling-based inference. This is achieved by optimizing each step in our proposed pipeline and can be considered a novelty by itself.  We will expand on these aspects to highlight the contribution of our work more clearly.
>
> > The method is not using minibatches, but instead, each step needs a full gradient, meaning that the cost of Bayesian neural networks is at least 2-3 orders of magnitude higher than using deterministic neural networks.
>
> **Limitation on Minibatching:** Exploring minibatch variants is indeed a promising future direction, which we made explicit in the paper’s discussion section. Our current focus, however, is on applications with manageable dataset sizes and advancing the scalability of full-batch samplers. Stochastic samplers come with various challenges that are worth studying on their own and would potentially dilute the performance gains of MILEs. In addition, various recent stochastic sampling approaches still struggle with being robust enough to be practically applicable without an extensive overhead in hyperparameter tuning [5-6].
>
> ---
> ### **Summary**
>
> We will add a discussion of Cobb et al. (2021) [3] to the related works section and add the benchmark study to the Appendix. In addition, we will incorporate the above points to make our contributions more clear. We believe that this will further strengthen the paper.  Please let us know if our response addresses your questions and concerns, and if there is anything else we need to clarify.
> Thank you again for your constructive feedback. Please let us know if our response addresses your questions and concerns, and if there is anything else we need to clarify.
>
> **References**
>
> [1] Robnik, J., & Seljak, U. (2024). Fluctuation without dissipation: Microcanonical Langevin Monte Carlo. In Symposium on Advances in Approximate Bayesian Inference (pp. 111–126). PMLR.
>
> [2] Cabezas, A., Corenflos, A., Lao, J., & Louf, R. (2024). BlackJAX: Composable Bayesian inference in JAX. arXiv.
>
> [3] Cobb, A. D., & Jalaian, B. (2021, December). Scaling Hamiltonian Monte Carlo inference for Bayesian neural networks with symmetric splitting. In Uncertainty in Artificial Intelligence (pp. 675-685). PMLR.
>
> [4] Sommer, E., Wimmer, L., Papamarkou, T., Bothmann, L., Bischl, B., & Rügamer, D. (2024). Connecting the Dots: Is Mode-Connectedness the Key to Feasible Sample-Based Inference in Bayesian Neural Networks? Proceedings of the 41st International Conference on Machine Learning, in Proceedings of Machine Learning Research 235:45988-46018.
>
> [5] Kim, S., Jung, S., Kim, S. ; Lee, J.. (2024). Learning to Explore for Stochastic Gradient MCMC. Proceedings of the 41st International Conference on Machine Learning, in Proceedings of Machine Learning Research 235:24015-24039
>
> [6] Yi-An Ma, Tianqi Chen, and Emily B. Fox. 2015. A complete recipe for stochastic gradient MCMC. In Proceedings of the 28th International Conference on Neural Information Processing Systems - Volume 2 (NIPS'15). MIT Press, Cambridge, MA, USA, 2917–2925.

---

> ### Author Response · Authors · 2024-11-23
>
> Dear Reviewer e8p6,
>
> Thank you for taking the time to read our response and providing further feedback. Please find below our response.
>
> > It could be that you did not notice this, but that paper is full of mistakes, such as the claim of Theorem 3 that the Euler-Mayurama discretization exactly preserves the invariant distribution is completely wrong.
>
> There might have been a misunderstanding. The theorem is correct, but maybe it was not clear what its claim is. The claim is that if you do Euler-Mayurama discretization of the stochastic-deterministic part of dynamics AND you were able to exactly solve both separate parts, then you would have no bias. In practice of course you do not know how to exactly solve the deterministic part, so you have to do further splitting to momentum and position updates which introduces bias. So the claim of the theorem is not that there is no bias, but that the stochastic part of the dynamics does not contribute to the bias. If you think that there are additional mistakes in that paper, please let us know. We will do our best to address them.
>
> > The code for the paper of Robnik and Seljak attempts to solve the SDE accurately by using an adaptive step size high order Runge-Kutte method,
>
> There might be another misunderstanding as this statement is also not correct as the deterministic part is solved by a splitting method, such as leapfrog or Omelyan integrator. It is not of the Runge-Kutte type and is not a higher order method (it is second order, as most integrators used in MCMC). While these explanations are stated in great detail in the “Microcanonical Hamiltonian Monte Carlo” paper by Robnik et al., we are happy to reiterate those in our paper in an additional section in the appendix.
>
> > I strongly feel that the precise statement of the discretization of the SDE should be included in the paper, otherwise it is absolutely unclear what is implemented in the code. You cannot expect readers to look at thousands of lines of code to figure out what is the key algorithm of the paper actually doing.
>
> We absolutely agree with the reviewer and are thankful for this suggestion. We will make this more clear in the updated version of the manuscript.
>
> > It is commendable that you attempted to compare the method with Symmetric Split HMC, but I am skeptical about the fact that the differences between the performance in terms of accuracy are so significant, as the Symmetric Split HMC has no bias (due to the Metropolis/Hastings step).
>
> > did not do any experiments to evaluate the bias of the method.
>
> As we explain in the text, having no bias at the cost of the Metropolis test is actually harmful for the performance of the algorithm, because it increases the variance. If the bias is small, but controlled, as it is in MCLMC by the energy error, the resulting algorithm achieves better performance. Our comparisons with BDEs (HMC-based sampling with correction step) confirm this hypothesis, showing that MCLMC indeed performs on par with (or sometimes even slightly better than) HMC. Experiments explicitly evaluating the bias of MCLMC are given in the respective paper “Microcanonical Hamiltonian Monte Carlo” by Robnik et al., see for example Figure 4.
>
> > With better tuning of the step size and the number of steps, as well as the batch size, you should be able to obtain similar accuracy
>
> We think that a useful sampling algorithm should work out of the box, with no need to additionally tune its hyperparameters on a problem-to-problem basis. This is exactly what we achieve with MILE and what we aimed to accomplish with our paper. As stated in our revision, Symmetric Split HMC can be a very useful approach and we agree that there might be hyperparameters for which a better performance than the one reported in our previous response can be obtained. In our above experiments, we used the hyperparameters provided by the authors as these already represent meaningful and tuned hyperparameters. However, as stated in our rebuttal, we had to limit sampling times to a certain budget to be able to respond within the rebuttal phase. We are more than happy to run Symmetric Split HMC including hyperparameter optimization and without budget constraint after the rebuttal phase to demonstrate the reviewer’s statement.
>
> ---
>
> ---
>
> Thank you again for your feedback. Please let us know if our response addresses your concerns, and if there is anything else we need to clarify.

---

> > ### Comment · Reviewer_e8p6 · 2024-11-24
> > **Response to official comment by authors**
> >
> > I remain unconvinced by the authors response, so I will keep my mark.
> > I hope that the authors can precisely state their algorithms in future submissions, which will allow them to make a clearer contrast with existing methods in the literature.

---

> > > ### Author Response · Authors · 2024-11-25
> > >
> > > Dear Reviewer e8p6,
> > >
> > > Thank you for your follow-up response, we know that your time is valuable. We would like to iterate all your initial concerns in order to make sure that we understand why you remain unconvinced.
> > >
> > > - Comparison with Split HMC: We have provided **additional experiments** following your request, **including the code to reproduce** these results. We used the best hyperparameters provided in the paper from Cobb&Jalaian, and would not dare to post wrong results on an open review platform. So it is unclear to us, where your skepticism stems from.
> > > - The reviewer’s skepticism about the method by Robnik et al. and its explanation in our work:
> > >     - We **clarified several misunderstandings** about the method and confirmed that Theorem 3 in Robnik et al. is correct.
> > >     - We made **significant efforts** to explain the discretization, bias, and comparisons.
> > >
> > >     It is not clear to us, what else the reviewer expects and where the remaining skepticism stems from.
> > >
> > > - Novelty: we clarified that our approach is not just putting ensembles together with MCLMC. Specifically,
> > >     - we found that MCLMC does not work for highly multimodal problems found in BNN applications, and previous HMC-based approaches as proposed in Sommer et al. (ICML 2024) are slow and exhibit unpredictable runtimes. Subsequently, we adapted MCLMC as a promising alternative to state-of-the-art NUTS.
> > >     - These adaptions involve a) combining MCLMC with an ensemble approach for better starting values, but also changing the originally proposed tuning phase, by modifying b) the step size, c) the energy variance scheduler, d) the effective sample size;
> > >     - furthermore, we e) eliminate a bottleneck in the originally proposed Phase III by replacing the FFT with a more efficient version.
> > >     - Our main contribution is thus
> > >          - a reliable, off-the-shelf approach for multimodal problems (see also new results provided in our previous response)
> > >         - for fast, tuning-free, and accurate sampling-based inference for BNNs (empirical evidence provided in the original submission and replies to your review).
> > >
> > >     Whether or not the reviewer finds this **novel** enough is a personal decision. These adaptations required a lot of work. Beating state-of-the-art samplers without additional tuning while being significantly faster is, if we may say so, in any case, a **significant** contribution to the community.
> > >
> > > **Summary**
> > >
> > > While we are very grateful for your initial constructive review, we struggle to find any clues about potentially unresolved points in your previous answers to our rebuttal, and on how we can further improve our paper beyond reiterating exactly the contents provided in the Robnik et al. paper. We also are clueless about why the Reviewer keeps the same mark and with a high confidence of 4 despite our answers providing additional empirical evidence (as requested), clarifying misunderstandings about the theory in Robnik et al., and despite reiterating the paper such that the reviewer’s initial concerns about the clarity can be addressed.
> > >
> > > Sincerely,
> > >
> > > The Authors

---

### Author Response · Authors · 2024-11-22
**Summary of Discussion and Key Updates – By the Authors**

**Dear Reviewers,**

As we approach the final quarter of the discussion phase, we wish to summarize the key points discussed and extend our gratitude to reviewer 83Bk for acknowledging our responses and increasing the score. Furthermore, we have submitted an improved manuscript based on your constructive comments and also included the additional code for the new experiments in the anonymous repository.

**Key Updates**

- **Additional comparisons**: Benchmarked MILE against symmetric Split HMC and PGPS, showcasing MILE’s superior speed and performance (Section 1, Appendix A.1.3).
- **Algorithm and Convergence Clarifications**: Detailed our algorithmic choices, in particular the one of unadjusted sampling and the resulting efficiency advantages (Sections 2, 3, Appendix A.3).
- **Highlight MILE’s Novelty**: We highlight MILE as a reliable off-the-shelf method for BNN inference, with critical adaptations for multimodality and robustness, as shown in a new benchmark (Section 2, Appendix A.1.2) where naive MCLMC fails up to 86% of runs while MILE achieves a failure rate of 0%. These novelties, alongside its unprecedented speed, and state-of-the-art performance, underline MILE’s practical appeal. In order to better convey this, we reformulated our contributions in Section 1, parts of Section 4 and the Discussion in Section 5.

We kindly invite the remaining reviewers to share their thoughts on whether our responses have adequately addressed their questions and concerns, as well as to let us know if there is anything else we should clarify or improve. Once again, we sincerely thank all reviewers for their valuable feedback, which has helped us enhance the paper by showcasing the strengths of our framework through additional benchmarks and more effectively highlighting our contributions.

---

### Author Response · Authors · 2024-12-02
**Final Reminder & Summary**

Dear Reviewers,

Thank you again for your feedback and engagement throughout this process. As the **discussion period closes in about 24 hours**, we would like to kindly remind you to share any final thoughts on our responses and the additional experiments provided.

To summarize:

- We addressed all concerns raised, providing clarifications on theoretical aspects, highlighting our empirical contributions, and demonstrating the robustness of MILE compared to existing methods.
- Misunderstandings about related work and our methodology were clarified, and we pointed to our comprehensive, anonymized code repository that supports reproducibility.
- Further experiments included during the rebuttal phase underline the practical significance of our approach, offering fast and accurate off-the-shelf sampling for BNNs, which is a significant contribution to the field.

We are confident that these responses fully address the concerns raised and hope they support a fair and constructive evaluation of the work.

Thank you for your time and consideration.

Sincerely,

The Authors

---

### Author Response · Authors · 2024-12-03
**Closing Summary by the Authors**

Dear Reviewers,

As the discussion phase concludes, we would like to summarize our method in one sentence to again highlight its significance and advantages:

***MILE** is a robust off-the-shelf method that generates high-quality posterior samples of BNNs in a fraction of the runtime of competing methods, delivering superior performance, in particular over sequential state-of-the-art samplers like NUTS and MCLMC.*

We have visualized this behavior in this **GIF**: [https://anonymous.4open.science/r/MILE-1CC1/README.md](https://anonymous.4open.science/r/MILE-1CC1/README.md), which is now included in the README of our repository.

We would further like to highlight that the same repository contains a fully reproducible setup of our experiments, including the benchmarks with new comparison methods as requested by Reviewer e8p6 and 83Bk.

We hope that we have eliminated all the remaining doubts and again thank all reviewers for their engagement and time investment.

Sincerely,

The Authors

---

### Meta-Review · Area_Chair_xXwW · 2024-12-20

**Metareview:**

This paper proposes MILE, an adaptation of the Microcanonical Langevin Monte Carlo (MCLMC) algorithm for sampling from Bayesian Neural Network (BNN) posteriors. The key claims are:

MILE achieves speedups compared to NUTS while maintaining or improving predictive performance
The method provides more reliable and predictable sampling through ensembling and careful tuning of MCLMC
MILE enables practical, scalable sampling-based inference for BNNs with runtime complexity comparable to deterministic optimization

Strengths

This is an BNN sampling is an importnat problem (IMO)
Extensive experimental validation on multiple datasets demonstrates improvements over existing methods
The authors provide thorough ablation studies showing the benefit of their adaptations to MCLMC
The work is well-presented with clear explanations of technical concepts

Weaknesses

Limited theoretical analysis of convergence properties
Some reviewers felt the novelty was incremental since it builds on MCLMC
The precise algorithmic details of the discretization scheme were  unclear
While claiming to be "tuning-free", the method still requires some parameter tuning, though automated
Full batch gradient computations are required

I recommend acceptance of this paper because:

It makes a meaningfully practical contribution to (almost) black-box BNN sampling
The empirical results are thorough, showing clear improvements over existing methods
The authors were responsive during rebuttal and provided additional experiments and clarifications
While building on existing work, the adaptations are non-trivial and necessary for making MCLMC work effectively for BNNs
I think the full-dataset gradient computation could be addressed in future work using existing techniques.

**Additional Comments On Reviewer Discussion:**

The review discussion centered around several areas:

Algorithmic Details & Theory

Reviewer e8p6 raised concerns about the precise discretization scheme and theoretical properties
The authors clarified misunderstandings about the theoretical claims in the original MCLMC paper
While more theoretical analysis would be nice, the empirical results are good

Novelty

Reviewers qCdd and k13v questioned the novelty beyond basic ensembling
Authors demonstrated through ablation studies that naive MCLMC fails most of the time without their adaptations
The significant speed improvements while maintaining accuracy seems to me like a meaningful contribution

Comparison with Other Methods

Reviewer 83Bk requested comparison with PGPS
Authors provided comprehensive benchmarks showing competitive or better performance with much faster runtimes
These results strengthened the paper's contributions

Reproducibility

Initially unclear algorithmic details were clarified
Authors provided access to code repository with all experiments
Additional benchmarks were conducted during rebuttal

The authors addressed most concerns constructively with additional experiments and clarifications. While some theoretical aspects could be strengthened, the practical significance and empirical validation make this a valuable contribution to the field.

One reviewer viewed the paper harshly. The authors commented to me saying that they were being somewhat hostile and seemingly judging them for failures of other scientists. While I think they were being a bit dramatic, I do agree somewhat with their characterization. I feel the authors did address the reviewer's concerns and the reviewer didn't seem to be listening. Because of this, I am inclined to discount their score. Of course, I still considered it, but I weighed it down in my head -- this down-weighting puts the paper's score into what I. feel is an acceptable range but I'm also alright is senior ACs disagree.

---

### Decision · Program_Chairs · 2025-01-22

Accept (Poster)